# AutoCodeBench: Large Language Models are Automatic Code Benchmark Generators

**Jason Chou**[*]   **Ao Liu**[*]   **Yuchi Deng**   **Zhiying Zeng**   **Tao Zhang**   **Haotian Zhu**
**Jianwei Cai**   **Yue Mao**   **Chenchen Zhang**   **Lingyun Tan**   **Ziyan Xu**   **Bohui Zhai**
**Hengyi Liu**   **Speed Zhu**   **Wiggin Zhou**[†]   **Fengzong Lian**[†]
Hunyuan Team, Tencent
`https://autocodebench.github.io/`

## Abstract

Large Language Models (LLMs) have shown impressive performance across diverse domains, with code generation emerging as a particularly prominent application. However, existing benchmarks designed to evaluate code generation exhibit several critical limitations. First, most rely on manual annotations, which are time-consuming and difficult to scale across programming languages and problem complexities. Second, the majority focus primarily on Python, while the few multilingual benchmarks suffer from limited difficulty and imbalanced language coverage. To overcome these challenges, we present **AutoCodeGen**, an automated framework for constructing high-difficulty, multilingual code generation datasets without manual annotations. Our approach guarantees correctness and completeness of test cases by generating test inputs with LLMs, obtaining test outputs within a multilingual sandbox, and further enhancing quality through reverse problem generation and multi-stage filtering. Based on this novel method, we introduce **AutoCodeBench**, a large-scale benchmark suite spanning 20 programming languages with balanced coverage. AutoCodeBench is designed to rigorously evaluate LLMs on diverse, challenging, and realistic multilingual programming tasks. Extensive experiments reveal that even state-of-the-art models struggle on these tasks, particularly in low-resource languages. Besides, we release complementary training and evaluation resources, including a large-scale, verifiable multilingual training set generated via the same pipeline, as well as a multilingual sandbox with high-concurrency support. We hope these contributions will provide a solid foundation for future research and inspire the community to explore more automatic and scalable approaches to multilingual code generation.

## 1 Introduction

Recently, Large Language Models (LLMs) have advanced rapidly, achieving strong performance across a wide range of tasks (OpenAI, 2024; Gemini, 2025; DeepSeek-AI, 2025b; Anthropic, 2025b). Among these, code generation has emerged as a central indicator of both intelligence and practical utility, drawing increasing attention from academia and industry alike (Chen et al., 2021; Jimenez et al., 2024; Jiang et al., 2024). Many powerful LLMs, such as Claude Opus 4.1 (Anthropic, 2025a), are already widely adopted in AI-assisted coding tools (Cursor, 2025; Anthropic, 2025). Through the ability to generate executable code, LLMs can substantially accelerate programming automation and reduce manual development effort.

To evaluate and advance these capabilities, a series of benchmarks have been developed (Wang et al., 2025a). Early efforts such as HumanEval (Chen et al., 2021) and MBPP (Austin et al., 2021) focused on short, algorithmic Python tasks. More recent benchmarks (Peng et al., 2024; Jimenez et al., 2024; Jain et al., 2025; Zheng et al., 2025; Zhu et al., 2025; Chai et al., 2025; Bytedance, 2025; Zan et al., 2025) target more challenging and realistic programming tasks, including

---

[*]Equal contributions.
[†]Project leads.

Table 1: Comparison of Code Generation Benchmarks. **MLing**: MultiLingual; **MLogi**: MultiLogical, refers to programming problems that require the model to simultaneously implement multiple core functionalities. **HFree**: Human-Free; **BDist**: Balanced Distribution of multiple languages. **DSize**: Data Size. **PLen**: Problem Length. Further details are provided in the Appendix B.

| Benchmark | MLing | MLogi | HFree | BDist | Difficulty | Category | DSize | PLen |
|---|---|---|---|---|---|---|---|---|
| HumanEval | ✘ | ✘ | ✘ | / | ★ | 5 | 164 | 134.1 |
| MBPP | ✘ | ✘ | ✘ | / | ★ | 6 | 378 | 50.5 |
| LiveCodeBench | ✘ | ✘ | ✘ | / | ★★★★ | 4 | 1100 | 469.6 |
| FullStackBench | ✔ | ✘ | ✘ | ✘ | ★★ | 12 | 1687 | 184.3 |
| McEval | ✔ | ✘ | ✘ | ✔ | ★★ | 9 | 2007 | 146.7 |
| **AutoCodeBench** | ✔ | ✔ | ✔ | ✔ | ★★★★ | **14** | **3920** | **498.2** |

competition-level problems and multilingual scenarios. However, constructing such benchmarks remains costly and labor-intensive, which hinders scalability and makes it difficult to guarantee both high difficulty and broad coverage. As shown in Table 1, widely used multilingual benchmarks such as FullStackBench (Bytedance, 2025) and McEval (Chai et al., 2025) suffer from imbalanced language distributions and limited diversity. These issues stem from the inherent bias of manual annotation. For example, most annotators are proficient in Python algorithmic tasks but lack expertise in domains such as Elixir-based communication development. Moreover, the rapid progress of LLMs has rendered many of the overly simple problems in these benchmarks obsolete. These limitations naturally raise a critical question: *Can we automatically construct high-quality code generation benchmarks that scale while ensuring both comprehensiveness and diversity?*

In this paper, we propose **AutoCodeGen**, an automated workflow centered on LLM–sandbox interaction, to synthesize challenging multilingual code generation datasets without manual annotations. Unlike previous data synthesis approaches (Luo et al., 2024; Wei et al., 2024b; Xu et al., 2025; Ahmad et al., 2025) that directly rely on LLMs to generate test functions, we adopt a more intuitive and reliable strategy. The LLMs first produce test inputs, and a multilingual sandbox executes these inputs to obtain the corresponding outputs. This design effectively mitigates a common limitation of LLMs, namely the tendency to generate incorrect outputs when confronted with high-difficulty problems. Furthermore, we introduce a reverse-generation paradigm, where code solutions and test functions are synthesized first, followed by the construction of the programming problem itself. This ensures that the resulting tasks are not only sufficiently challenging but also verifiably correct.

Based on the automation workflow, we introduce **AutoCodeBench**, a large-scale, fully automated code generation benchmark, as shown in Table 1. Compared with previous multilingual benchmarks (Cassano et al., 2022; Bytedance, 2025; Chai et al., 2025), ours simultaneously offers high difficulty, diversity, and practicality, with a balanced distribution of problems across 20 programming languages. We intentionally include some multi-logical problems to test the LLMs' capacity for multitasking in a single problem. The key contributions of this paper are as follows:

1. **AutoCodeGen**. We propose an automated workflow based on LLM-sandbox interaction, where LLMs generate test inputs and a sandbox executes them to obtain outputs, to create high-quality multilingual code generation tasks.

2. **AutoCodeBench**. We introduce AutoCodeBench, a large-scale, fully automatic code generation benchmark with 3,920 problems, evenly distributed across 20 programming languages, featuring high difficulty, practicality, and diversity. We also construct Lite and Complete versions to enable efficient and high-quality evaluation. The evaluation results show that current LLMs still struggle with complex and diverse multilingual programming tasks, especially in multi-logical and low-resource language scenarios.

3. **Training and Evaluation Resources**. We propose **AutoCodeInstruct**, a multilingual code generation training set constructed using the same pipeline as AutoCodeBench, ensuring comparable quality. We design a two-stage GRPO training to demonstrate the potential of this dataset in enhancing code generation capabilities. Besides, we release a **Multilingual**

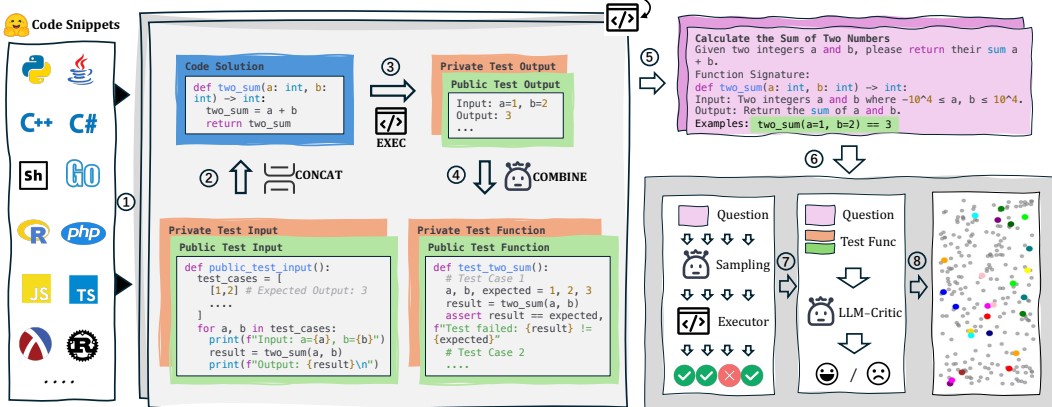

Figure 1: **The overview of AutoCodeGen**. It first generates code solution and the corresponding public/private test input functions based on multilingual code snippets (①). They are concatenated and executed in a sandbox to obtain test outputs, which are then combined by the LLM into complete test functions (②,③,④). Based on the code solution and test function, the LLM is prompted to generate accurate programming problems (⑤). Finally, a three-stage data filtering is applied: multiple sampling to remove too easy problems (⑥), LLM-as-Critic to discard low-quality ones (⑦), and diversity-based tagging to ensure distributional variety (⑧).

**Sandbox** with high-concurrency support, which can be employed for both model evaluation and RL training.

## 2 AUTOCODEGEN & AUTOCODEBENCH

In this section, we first present how AutoCodeGen constructs the AutoCodeBench family of benchmarks, and then provide an overview of AutoCodeBench.

### 2.1 AUTOCODEGEN

Our AutoCodeGen is a fully automated workflow based on LLM-sandbox interaction for constructing verifiable code generation datasets. It first generates large-scale multilingual data with guaranteed executability and correctness, then applies a three-stage filtering strategy to ensure the benchmark is challenging, high-quality, and diverse. As illustrated in Figure 1, the workflow includes four key stages: Code Solution Generation (①), Test Function Generation (②,③,④), Programming Problem Generation (⑤), and Data Filtering (⑥,⑦,⑧).

#### 2.1.1 CODE SOLUTION GENERATION

We begin by extracting multilingual code snippets from Stack-Edu (Allal et al., 2025), a large-scale dataset of educational code filtered from The Stack v2 (Lozhkov et al., 2024), as seeds. These seeds span function-level, class-level, and file-level code, sourced from real GitHub repositories, ensuring diversity and practicality. Using a language-specific few-shot prompt, we guide `DeepSeek-V3-0324` to refine and evolve these seeds into verifiable and self-contained code solutions. During this process, the model removes non-essential logic and adds appropriate comments for clarity. We then validate the correctness of the generated solutions by multilingual sandbox.

#### 2.1.2 TEST FUNCTION GENERATION

We enhance efficiency and edge-case coverage by first generating test inputs via LLMs and then executing them in a sandbox to obtain the corresponding outputs. Specifically, it is divided into the following three steps:

**Test Input Generation** The test input functions (both public and private) are generated alongside the above code solution, ensuring alignment between the solution and its inputs. The public test input

Table 2: Programming language translation pairs.

| Origin | Target | Origin | Target | Origin | Target | Origin | Target | Origin | Target |
|--------|--------|--------|--------|--------|--------|------------|-----------|--------|--------|
| Python | R | Python | Ruby | Java | Scala | Java | C# | Shell | Perl |
| Python | Elixir | Python | Julia | Java | Kotlin | JavaScript | PHP | C++ | Rust |
| Python | Swift | Python | Racket | Java | Dart | JavaScript | Typescript | | |

function includes no more than 3 basic cases and serves demonstration purposes; it will be embedded into the final programming problem as an illustrative usage. In contrast, the private test input function contains 7+ cases, including edge cases, and functions as the comprehensive test for verifying the correctness of the code solution.

**Test Output Generation** We concatenate the code solution with test input functions and execute them in the sandbox to obtain the corresponding test outputs.

**Input-Output Integration** We prompt `DeepSeek-V3-0324` with both the test input functions and output results from the sandbox to generate coherent and verifiable test functions. Finally, we validate the correctness by executing the code solution together with the generated public and private test functions in the sandbox.

### 2.1.3 Programming Problem Generation

Generating high-quality programming problems is challenging, as it requires detailed and accurate problem descriptions. We find that models often omit key information when generating programming problems, such as the entry point specified in the test function. Therefore, we define a set of specifications, such as explicit input/output formats, function and class names, to ensure that the generated problems are detailed and well-structured. We prompt `DeepSeek-V3-0324` to generate high-quality programming problems based on the code solution with appropriate comments and the corresponding test function, while embedding the public test function as example usage.

**Through these three steps, we obtain a large-scale multilingual dataset, where each instance is represented as a tuple <programming problem, code solution, public test function, private test function>.**

### 2.1.4 Data Filtering

Finally, We apply three filtering and sampling steps to ensure the high-difficulty, high-quality, and diversity of the final benchmark.

**Difficulty Control** Too simple programming problems are barely meaningful for evaluating the code generation capabilities of current LLMs. To address this, we employ a moderately capable code model, `DeepSeek-Coder-V2-Lite`, to filter out too easy problems. Specifically, we sample answers for each problem ten times using the model and validate the correctness via sandbox execution. We discard problems that are solved in all attempts. Take Python as an example, `DeepSeek-Coder-V2-Lite` can filter out 25.1% of the whole problems.

**Quality Control** During the aforementioned problem generation stage, we define six specifications to guide the generation of detailed and accurate programming problems. To further ensure high quality, we employ `DeepSeek-R1-0528` to critique each <problem, test function> pair. Only the data whose test functions are completely accurate and fully aligned with the problem are retained.

**Diversity Sampling** We aim for our benchmark to cover as many real-world scenarios as possible. To this end, we perform diversity-based sampling on the existing data to construct the final benchmark. We use `DeepSeek-V3-0324` to label each problem. We then divide the problems into different pools by category and perform cyclic sampling, ensuring a broad representation of programming scenarios.

### 2.1.5 Approximate Language Translation

For Python, C++, Shell, Java, JavaScript, and Go, we directly use the workflow described above. For the other 14 languages, while the proposed workflow is still applicable, we choose to employ an

Table 3: Statistics of Dataset. **ACB**: AutoCodeBench; **Langs**: Languages; **Prob**: Problem; **Solu**: Solution; **Len**: Length; **E/M/H**: Easy/Medium/Hard. The difficulty level is determined by the number of passes in ten samplings of `DeepSeek-Coder-V2-Lite`. Problems with zero correct solutions are classified as hard, 1-5 correct solutions as medium, and those with more than five as easy.

|  | #Problems | #Test Cases | #Langs | Prob Len | Solu Len | Difficulty (E/M/H) |
|---|---|---|---|---|---|---|
| ACB-Full | 3,920 | 37,777 | 20 | 498.2 | 487.5 | 646/846/2428 |
| ACB-Lite | 1,586 | 15,341 | 20 | 517.2 | 469.3 | 263/421/902 |
| ACB-Complete | 1,000 | 9,608 | 20 | 505.2 | 461.2 | 169/265/566 |

approximate language translation approach due to their limited data resources and lack of diversity. We extract unused data from the dataset generated in Section 2.1.3 and translate them into the target low-resource language, as shown in Table 2. This ensures a sufficient and diverse dataset, which is further refined through the Data Filtering process in Section 2.1.4.

## 2.2 AUTOCODEBENCH

### 2.2.1 DATA OVERVIEW

As shown in Table 1 and 3, AutoCodeBench(-Full) is a large-scale, high-difficulty multilingual benchmark. Over 60% of the problems are classified as hard problems, with each problem averaging 498.2 characters and accompanied by 9.6 test cases, providing a challenging and comprehensive evaluation standard. The 20 languages are as follows: *Python, C++, Java, JavaScript(JS), Go, Shell, C#, Dart, Elixir, Julia, Kotlin, Perl, PHP, Racket, R, Ruby, Rust, Scala, Swift, TypeScript(TS)*.

To analyze the diversity and language coverage of AutoCodeBench, we first use `Claude Sonnet 4` to generate 20 language-agnostic task categories, and then employ `DeepSeek-V3-0324` to classify each problem accordingly. Categories with less than 2% representation are merged into the "Other" group. AutoCodeBench covers 14 categories, demonstrating comprehensive coverage of practical programming scenarios. Besides, we analyze the distribution of problems across the 20 programming languages. AutoCodeBench exhibits a relatively balanced distribution across languages, with no significant bias toward any specific one, further validating its completeness and representativeness as a multilingual benchmark. Detailed category and language distribution are provided in Appendix B.3.

### 2.2.2 AUTOCODEBENCH-LITE AND AUTOCODEBENCH-COMPLETE CONSTRUCTION

To facilitate quicker and more efficient model evaluations, we create **AutoCodeBench-Lite**, a simplified subset of AutoCodeBench. Specifically, we collect the problem-solving results from all models and sort the problems in ascending order based on the number of passes. After discarding problems with fewer than 2 passes, we select approximately 1,500 problems based on their pass count in ascending order. These problems, which have been solved correctly by existing models at least twice and have a certain level of difficulty, are selected to amplify the differences between the models. We use these problems as the set for the Lite version.

To enable evaluating base models, we further present **AutoCodeBench-Complete**, a completion-based version of ACB. Concretely, we select 1,000 data points from ACB-Lite to ensure a balanced distribution of 50 problems per programming language and use 3-shot demonstrations to evaluate the performance of base models. ACB-Complete can serve as a comprehensive benchmark for evaluating the multilingual code generation capabilities of base models.

## 3 EVALUATION

### 3.1 EVALUATION SETUP

We use the Pass@1 (%) (Chen et al., 2021) as the default evaluation metric. In terms of inference parameters, for proprietary models and open-source models that provide APIs, we access them through direct API calls. Other models are evaluated with official parameters when available, or

| | Average | Python | Cpp | Java | JS | Go | Shell | Csharp | Dart | Elixir | Julia | Kotlin | Perl | PHP | Racket | R | Ruby | Rust | Scala | Swift | TS |
|---|---|---|---|---|---|---|---|---|---|---|---|---|---|---|---|---|---|---|---|---|---|
| Count | 3920 | 196 | 186 | 188 | 184 | 191 | 188 | 199 | 200 | 198 | 200 | 200 | 200 | 199 | 196 | 198 | 200 | 199 | 199 | 200 | 199 |
| *Current Upper Bound* | *75.3* | *65.3* | *75.8* | *80.9* | *60.9* | *71.7* | *72.9* | *88.4* | *78.0* | *97.5* | *78.5* | *90.5* | *64.5* | *53.8* | *88.9* | *75.8* | *81.0* | *62.8* | *78.4* | *78.5* | *61.3* |
| **Proprietary Models and 200B+ Open-source Models** | | | | | | | | | | | | | | | | | | | | | |
| Claude Opus 4.1 (20250805) | 55.4 | 42.3 | 49.5 | 56.4 | 42.9 | 44.0 | 50.0 | 78.4 | 59.5 | 86.9 | 59.5 | 74.5 | 47.5 | 31.2 | 73.0 | 55.6 | 61.0 | 39.2 | 50.8 | 57.0 | 47.7 |
| Claude Sonnet 4 (20250514) | 51.1 | 37.2 | 46.8 | 52.7 | 34.8 | 41.9 | 48.9 | 72.4 | 53.5 | 81.8 | 49.0 | 71.5 | 45.0 | 34.7 | 68.9 | 50.5 | 54.5 | 36.2 | 48.2 | 48.0 | 44.2 |
| Claude Opus 4.1 (20250805) | 52.6 | 38.3 | 48.4 | 53.7 | 41.3 | 38.7 | 46.8 | 75.4 | 55.0 | 80.3 | 57.5 | 76.0 | 45.0 | 29.6 | 64.8 | 51.0 | 55.5 | 39.7 | 51.8 | 53.0 | 47.2 |
| Claude Sonnet 4 (20250514) | 49.3 | 35.7 | 47.3 | 52.7 | 38.0 | 37.7 | 47.9 | 72.9 | 51.0 | 74.2 | 51.0 | 72.0 | 44.0 | 30.7 | 63.8 | 44.4 | 51.5 | 35.2 | 45.2 | 45.5 | 44.2 |
| GPT-5 (20250807) | 53.5 | 44.4 | 51.6 | 48.9 | 44.6 | 45.0 | 47.3 | 75.9 | 53.5 | 84.3 | 53.0 | 70.5 | 43.7 | 36.2 | 58.5 | 54.0 | 60.5 | 40.7 | 49.7 | 58.5 | 46.7 |
| o3-high (20250416) | 51.1 | 40.8 | 47.3 | 53.2 | 40.8 | 22.0 | 49.5 | 68.3 | 55.0 | 80.8 | 54.5 | 72.0 | 44.0 | 32.7 | 53.1 | 47.5 | 59.0 | 42.2 | 51.3 | 59.0 | 47.2 |
| o4-mini (2025-04-16) | 50.0 | 42.3 | 46.8 | 51.6 | 40.2 | 31.4 | 45.2 | 68.3 | 54.0 | 82.3 | 49.0 | 74.0 | 44.0 | 30.2 | 45.4 | 43.4 | 59.0 | 40.2 | 50.3 | 54.0 | 45.7 |
| GPT4.1 (2025-04-14) | 48.0 | 37.2 | 46.8 | 48.9 | 34.8 | 37.2 | 36.7 | 74.4 | 46.5 | 76.8 | 50.0 | 72.0 | 43.5 | 29.2 | 50.5 | 42.4 | 54.0 | 37.2 | 44.2 | 49.5 | 46.2 |
| Grok-4 | 50.9 | 41.2 | 48.7 | 50.0 | 37.5 | 41.4 | 47.3 | 72.4 | 49.5 | 76.8 | 55.0 | 70.0 | 44.0 | 27.1 | 63.8 | 48.5 | 61.5 | 37.7 | 52.8 | 51.5 | 40.7 |
| Gemini2.5 Pro | 48.7 | 40.3 | 47.5 | 53.2 | 37.0 | 37.2 | 45.2 | 70.9 | 54.0 | 68.7 | 54.0 | 72.0 | 41.0 | 29.7 | 52.6 | 49.5 | 56.5 | 24.6 | 46.7 | 49.5 | 41.7 |
| Gemini2.5 Flash | 45.7 | 39.3 | 44.1 | 50.0 | 33.2 | 33.0 | 37.8 | 68.3 | 49.5 | 64.0 | 47.5 | 70.0 | 39.5 | 24.1 | 38.3 | 51.5 | 53.0 | 36.2 | 44.2 | 46.5 | 41.2 |
| DeepSeek-V3.1-250821 | 48.2 | 39.3 | 47.3 | 53.7 | 37.0 | 30.4 | 38.3 | 71.9 | 49.5 | 75.8 | 53.0 | 67.5 | 43.0 | 29.6 | 52.6 | 48.0 | 54.0 | 39.2 | 45.2 | 49.0 | 38.2 |
| DeepSeek-V3.1-250821 | 46.2 | 35.7 | 44.1 | 54.3 | 35.3 | 29.3 | 36.2 | 68.3 | 44.0 | 72.7 | 49.5 | 64.5 | 44.0 | 29.6 | 52.6 | 47.0 | 51.0 | 33.2 | 42.2 | 47.0 | 42.7 |
| DeepSeek-Coder-V2-Instruct | 37.7 | 29.1 | 34.9 | 34.0 | 27.7 | 29.8 | 31.4 | 63.8 | 33.5 | 60.6 | 37.5 | 58.5 | 35.5 | 25.1 | 41.8 | 35.4 | 45.0 | 22.6 | 33.2 | 38.0 | 35.7 |
| Hunyuan-TurboS-20250716 | 43.8 | 34.2 | 34.9 | 47.9 | 32.6 | 34.6 | 38.3 | 64.8 | 44.5 | 70.7 | 47.0 | 62.0 | 42.0 | 30.2 | 45.9 | 39.9 | 53.0 | 30.7 | 39.2 | 39.5 | 42.2 |
| GLM-4.5-enable | 46.6 | 41.0 | 43.2 | 47.9 | 34.8 | 37.8 | 43.9 | 70.5 | 42.0 | 72.5 | 47.5 | 66.0 | 43.5 | 28.6 | 50.0 | 45.0 | 54.5 | 31.6 | 41.0 | 46.0 | 42.2 |
| Kimi-K2-0905-preview | 46.8 | 36.2 | 38.2 | 47.3 | 37.0 | 35.1 | 41.5 | 68.3 | 50.5 | 78.8 | 48.5 | 66.5 | 41.5 | 30.7 | 55.6 | 40.4 | 49.5 | 31.7 | 45.7 | 48.5 | 42.7 |
| ERNIE-X1-Turbo-32K | 39.6 | 39.4 | 17.8 | 33.2 | 32.6 | 37.4 | 33.9 | 46.0 | 33.0 | 68.9 | 54.0 | 49.5 | 39.5 | 23.9 | 45.3 | 44.3 | 48.0 | 20.8 | 40.4 | 44.0 | 37.7 |
| Qwen3-235B-A22B-Thinking-2507 | 47.7 | 37.8 | 41.9 | 48.4 | 39.7 | 39.8 | 45.2 | 71.9 | 46.0 | 79.8 | 48.5 | 58.0 | 40.5 | 29.1 | 56.6 | 49.0 | 55.0 | 35.7 | 40.4 | 46.0 | 44.2 |
| Qwen3-Coder-480B-A35B-Instruct | 44.8 | 39.4 | 41.1 | 51.1 | 27.9 | 31.4 | 41.1 | 63.0 | 36.5 | 73.7 | 49.5 | 63.1 | 41.0 | 27.2 | 56.3 | 42.7 | 51.5 | 25.4 | 42.1 | 47.5 | 41.9 |
| Qwen3-235B-A22B-Instruct-2507 | 43.1 | 35.7 | 38.2 | 49.5 | 29.3 | 33.5 | 40.4 | 67.3 | 39.5 | 59.1 | 46.0 | 59.5 | 44.5 | 26.1 | 49.5 | 44.0 | 46.5 | 24.6 | 37.7 | 46.0 | 43.2 |
| Seed1.6-Thinking-250715 | 45.0 | 40.3 | 45.2 | 50.0 | 33.2 | 38.2 | 39.9 | 67.3 | 36.5 | 67.7 | 51.0 | 61.0 | 41.0 | 26.1 | 51.0 | 44.9 | 55.5 | 27.6 | 37.2 | 46.5 | 38.7 |
| Seed1.6-enabled (250615) | 45.3 | 39.8 | 44.6 | 46.3 | 28.3 | 40.8 | 44.1 | 60.3 | 39.5 | 69.7 | 51.0 | 58.0 | 41.5 | 25.6 | 52.6 | 51.0 | 52.0 | 28.6 | 41.7 | 47.5 | 41.2 |
| Seed1.6-disabled (250615) | 42.9 | 35.2 | 40.3 | 46.8 | 32.6 | 34.6 | 35.1 | 70.9 | 42.5 | 69.7 | 45.0 | 62.0 | 39.5 | 23.1 | 49.5 | 40.4 | 46.5 | 28.1 | 32.7 | 40.0 | 42.7 |
| **Open-source Models below 200B** | | | | | | | | | | | | | | | | | | | | | |
| GLM-4.5-Air-enable | 40.8 | 39.3 | 37.6 | 39.4 | 31.0 | 39.8 | 36.7 | 66.3 | 38.0 | 61.5 | 42.0 | 53.0 | 40.5 | 27.1 | 40.3 | 39.0 | 47.0 | 25.1 | 30.5 | 38.5 | 42.7 |
| Qwen3-Next-80B-A3B-Thinking | 40.6 | 38.3 | 39.8 | 43.6 | 38.0 | 33.0 | 37.4 | 66.3 | 24.0 | 59.1 | 43.5 | 43.5 | 42.5 | 25.1 | 34.2 | 46.0 | 50.5 | 26.1 | 35.2 | 43.7 | 41.2 |
| Qwen3-Next-80B-A3B-Instruct | 39.6 | 36.7 | 35.5 | 44.1 | 32.6 | 29.8 | 39.9 | 62.8 | 35.5 | 63.1 | 39.0 | 41.0 | 42.0 | 27.6 | 41.3 | 39.4 | 39.0 | 24.6 | 34.7 | 42.0 | 40.2 |
| Qwen3-32B | **41.7** | 37.8 | 38.7 | 39.9 | 32.6 | 36.1 | 39.4 | 67.8 | 34.5 | 65.2 | 42.5 | 52.0 | 40.5 | 27.6 | 37.8 | 44.9 | 47.0 | 28.1 | 37.2 | 42.0 | 40.2 |
| Qwen3-14B | 37.6 | 37.8 | 35.5 | 35.1 | 30.4 | 30.4 | 36.2 | 60.8 | 29.0 | 62.1 | 34.5 | 44.5 | 37.5 | 23.1 | 44.9 | 36.9 | 43.5 | 24.6 | 28.6 | 36.0 | 38.7 |
| Qwen3-8B | 28.5 | 28.1 | 22.6 | 21.8 | 28.3 | 29.3 | 27.1 | 52.8 | 21.0 | 43.9 | 29.0 | 36.0 | 35.5 | 18.6 | 13.3 | 30.8 | 37.0 | 12.6 | 21.1 | 22.0 | 37.7 |
| Qwen3-1.7B | 11.2 | 16.8 | 5.4 | 4.8 | 12.5 | 9.9 | 11.7 | 19.6 | 7.0 | 20.7 | 11.0 | 9.0 | 19.5 | 7.5 | 5.6 | 9.6 | 21.0 | 0.0 | 2.5 | 10.0 | 19.6 |
| Qwen3-32B | 31.0 | 26.5 | 21.5 | 29.5 | 28.0 | 25.5 | 24.0 | 59.3 | 27.5 | 52.0 | 28.0 | 45.0 | 34.5 | 21.6 | 28.6 | 22.7 | 36.5 | 16.1 | 26.6 | 30.0 | 35.2 |
| Qwen3-14B | 28.6 | 24.5 | 22.6 | 32.4 | 27.2 | 16.8 | 23.9 | 50.8 | 21.0 | 42.4 | 22.5 | 42.0 | 34.5 | 24.6 | 28.1 | 26.3 | 33.5 | 17.1 | 20.6 | 26.5 | 32.7 |
| Qwen3-8B | 23.3 | 22.4 | 11.3 | 25.0 | 22.8 | 18.3 | 22.3 | 42.2 | 17.0 | 41.4 | 18.5 | 36.0 | 29.5 | 18.1 | 23.5 | 16.2 | 27.0 | 7.5 | 19.1 | 17.5 | 29.1 |
| Qwen3-1.7B | 7.9 | 8.7 | 1.1 | 2.7 | 8.2 | 3.7 | 11.7 | 9.5 | 3.5 | 17.2 | 6.0 | 11.5 | 15.5 | 7.5 | 4.6 | 7.6 | 14.5 | 0.5 | 3.0 | 6.0 | 14.6 |
| Qwen2.5-Coder-32B-Instruct | 35.8 | 29.6 | 27.4 | 33.0 | 29.9 | 23.0 | 29.3 | 58.3 | 34.5 | 59.6 | 35.5 | 56.0 | 38.5 | 26.1 | 35.7 | 31.3 | 40.0 | 23.1 | 29.6 | 39.0 | 35.2 |
| Qwen2.5-Coder-7B-Instruct | 22.5 | 19.9 | 8.6 | 22.3 | 21.2 | 12.6 | 21.3 | 38.7 | 18.5 | 47.0 | 18.0 | 39.0 | 27.5 | 15.1 | 24.0 | 17.7 | 29.5 | 7.0 | 19.1 | 17.0 | 24.6 |
| Qwen2.5-Coder-1.5B-Instruct | 10.3 | 12.2 | 2.7 | 4.8 | 12.0 | 7.3 | 14.4 | 17.6 | 6.5 | 35.4 | 4.0 | 16.5 | 15.0 | 5.0 | 7.1 | 7.6 | 15.5 | 1.0 | 4.5 | 5.0 | 11.1 |
| DeepSeek-Coder-33B-Instruct | 28.5 | 25.0 | 24.2 | 29.3 | 24.5 | 29.8 | 22.3 | 54.8 | 17.5 | 67.7 | 14.5 | 52.0 | 29.5 | 19.1 | 28.1 | 18.7 | 33.0 | 8.0 | 24.1 | 18.0 | 29.1 |
| DeepSeek-Coder-6.7B-Instruct | 20.5 | 18.9 | 12.9 | 19.7 | 19.6 | 21.5 | 16.0 | 44.2 | 11.5 | 47.5 | 15.5 | 45.5 | 21.5 | 10.6 | 15.3 | 13.1 | 27.5 | 6.0 | 11.1 | 8.0 | 23.6 |
| Seed-Coder-8B-Instruct | 32.3 | 23.5 | 23.7 | 33.5 | 28.8 | 22.5 | 20.7 | 54.8 | 30.5 | 57.1 | 33.0 | 52.5 | 34.0 | 25.1 | 36.7 | 28.3 | 35.5 | 15.6 | 29.6 | 28.0 | 31.2 |
| OpenCoder-8B-Instruct | 19.3 | 14.3 | 9.7 | 19.1 | 12.5 | 17.3 | 21.3 | 33.2 | 15.5 | 34.3 | 15.5 | 30.0 | 26.5 | 14.6 | 15.8 | 17.2 | 29.5 | 6.0 | 17.1 | 15.0 | 21.1 |

Table 4: Pass@1 (%) performance of different models for AutoCodeBench. *Current Upper Bound* represents the Pass@1 value calculated by taking the union of problems correctly solved by all models. **Blue** and **Green** denotes reasoning and non-reasoning modes.

with greedy decoding otherwise. All models are provided with our custom system prompt, which standardizes the output format: ***You are an expert programmer. Your task is to provide a code solution within a single Markdown code block for the given programming problem. Do not include any direct execution commands, test cases, or usage examples within the code block.***

## 3.2 MAIN RESULTS

We comprehensively evaluate the performance on ACB, with results across different programming languages shown in Tables 4. The results of ACB-Lite and leaderboards are shown in Table 9, Figure 7 and Figure 8.

Results show that ACB is highly challenging, as no model surpasses 55.5 average score, indicating that current LLMs still struggle with complex, practical multilingual problems. Among all models, `Claude Opus 4.1` consistently achieves the best performance in both reasoning and non-reasoning modes, confirming its strength across diverse coding tasks and aligning with observations from SWE-bench (Jimenez et al., 2024). Finally, while individual models perform moderately, their combined upper bound reaches 75.3, revealing complementary strengths and substantial room for improvement, as no single model dominates across all languages.

## 3.3 PERFORMANCE ACROSS POPULAR AND LOW-RESOURCE PROGRAMMING LANGUAGES

We select five models with similar performance levels (ranging from 47.7 to 49.3) and evaluate their performance differences across popular and low-resource languages. As shown in Figure 2, the difference in average Pass@1 scores among the models for popular languages is small ($\Delta$ 3.1). However, the performance gap between models widens ($\Delta$6.3) in low-resource languages, suggesting that low-resource programming languages may have received insufficient attention in model development. Besides, since we use the moderately capable `DeepSeek-Coder-V2-Lite` as a filter to remove simple problems, the Pass@1 scores of top models on popular languages are relatively low. However, because the filter itself performs poorly on low-resource languages, many problems that appear trivial to top models are not filtered out, resulting in higher Pass@1 scores in these languages than in popular ones. This further highlights the pronounced disparities in

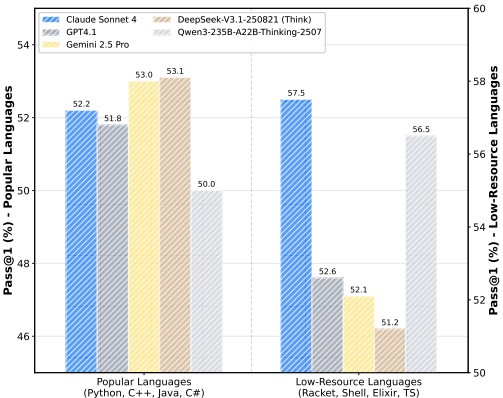 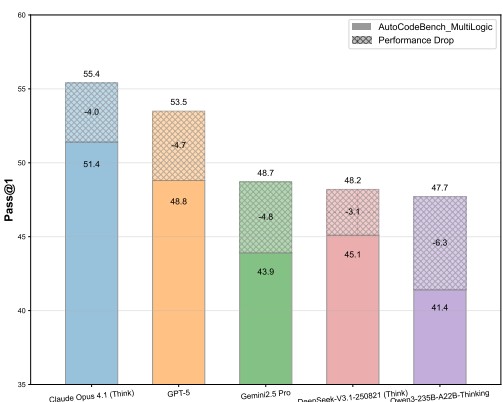

Figure 2: The performance comparison of different models across two language sets.

Figure 3: Performance drop of models on multi-logic problems (1,622) compared to full dataset.

low-resource language capabilities across models and underscores the need for greater community attention to this issue.

### 3.4 PERFORMANCE ACROSS MULTI-LOGIC PROGRAMMING PROBLEMS

A key feature that distinguishes AutoCodeBench from prior benchmarks is the inclusion of multi-logical problems. These problems require models to implement multiple distinct functions or classes within a single task, challenging their ability to handle multiple core demands simultaneously. We use `DeepSeek-V3-0324` to identify all multi-logical problems in AutoCodeBench and evaluate model performance on them. The results, shown in Figure 3, reveal a significant performance drop for all models when faced with multi-logical tasks. Among them, `Claude Opus 4.1` and `DeepSeek-V3.1` exhibit relatively smaller declines, while the other models show larger drops. These findings highlight a key limitation: current models still struggle with multi-logical problem solving, an ability that is particularly critical for real-world code agent applications.

### 3.5 PERFORMANCE ANALYSIS OF MULTI-TURN REFINEMENT WITH SANDBOX FEEDBACK

As shown in Figure 4, we evaluate how models leverage execution error messages to iteratively refine their code solutions. The results highlight the substantial value of our multilingual sandbox error feedback across all evaluated models. `Qwen2.5-Coder-32B-Instruct` achieves remarkable improvement from 35.8% to 47.4% after three refinement turns, while `Qwen3-8B` shows consistent progress from 23.3% to 30.2%. The most significant performance gains occur during the first refinement turn, with diminishing returns in subsequent iterations. This pattern suggests that models can effectively leverage execution feedback to identify and correct common coding errors, though the complexity of remaining problems increases with each iteration. The consistent improvement across different model scales indicates that multi-turn refinement with sandbox feedback is a valid strategy for enhancing code generation quality.

### 3.6 AUTOCODEBENCH-COMPLETE: EVALUATING BASE MODEL CAPABILITIES

Table 5 presents a performance comparison between base models on ACB-Complete and chat models on ACB-Full. Among models with 8B parameters or fewer, `Seed-Coder-8B` demonstrates superior performance in ACB-Complete, consistent with its strong showing on ACB-Full. This consistency suggests that the pretraining process effectively equipped `Seed-Coder-8B` models with strong multilingual programming capabilities, enabling them to handle diverse coding scenarios across multiple languages. Besides, an interesting observation arises when comparing `Qwen2.5-Coder-7B` and `OpenCoder-8B`. While it outperforms `OpenCoder-8B` on ACB-Full, the trend reverses on ACB-Complete. This suggests that `Qwen2.5-Coder-7B` may have undergone more effective post-training on multilingual code generation data.

Table 5: The pass@1 values of chat models (ACB-Full) and base models (ACB-Complete).

|  | ACB-Full | ACB-Complete |
|---|---|---|
| **30B+ Models** | | |
| DeepSeek-Coder-V2 | 37.7 | 39.0 |
| Qwen2.5-72B | 34.3 | 35.9 |
| Qwen2.5-Coder-32B | 35.8 | 35.5 |
| **~8B Models** | | |
| Seed-Coder-8B | 32.3 | 31.6 |
| Qwen3-8B | 23.3 | 22.6 |
| OpenCoder-8B | 19.3 | 26.1 |
| Qwen2.5-Coder-7B | 22.5 | 24.6 |
| DeepSeek-Coder-6.7B | 20.5 | 22.9 |

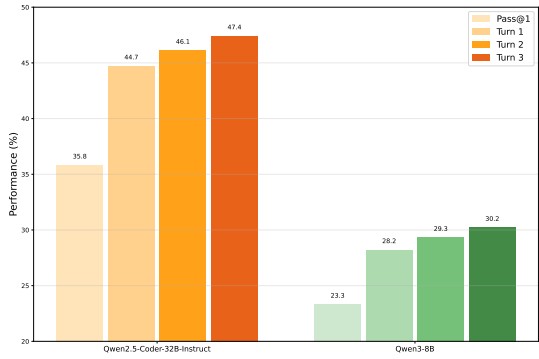

Figure 4: Performance improvement across multi-turn refinement with sandbox feedback.

Table 6: Results of two-stage GRPO and SFT with AutoCodeInstruct.

| Model | ACB-Full | ACB-Lite | LiveCodeBench-V6 | FullStackBench | McEval |
|---|---|---|---|---|---|
| Qwen2.5-Coder-7B-Instruct | 22.5 | 21.5 | 18.3 | 41.1 | 57.2 |
| + first-stage GRPO | $25.0_{\uparrow 2.5}$ | $24.8_{\uparrow 3.3}$ | $18.3_{\uparrow 0.0}$ | $46.9_{\uparrow 5.8}$ | $58.6_{\uparrow 1.4}$ |
| + second-stage GRPO | $27.4_{\uparrow 4.9}$ | $27.6_{\uparrow 6.1}$ | $17.1_{\downarrow 1.2}$ | $47.7_{\uparrow 6.6}$ | $58.4_{\uparrow 1.2}$ |
| + SFT | $28.9_{\uparrow 6.4}$ | $29.0_{\uparrow 6.5}$ | $17.7_{\downarrow 0.6}$ | $47.7_{\uparrow 6.6}$ | $63.1_{\uparrow 5.9}$ |
| Qwen2.5-Coder-32B-Instruct | 35.8 | 37.4 | 24.0 | 57.1 | 64.5 |
| + first-stage GRPO | $38.3_{\uparrow 2.5}$ | $39.5_{\uparrow 2.1}$ | $25.1_{\uparrow 1.1}$ | $58.3_{\uparrow 1.2}$ | $65.4_{\uparrow 0.9}$ |
| + second-stage GRPO | $41.6_{\uparrow 5.8}$ | $45.3_{\uparrow 7.9}$ | $28.0_{\uparrow 4.0}$ | $59.7_{\uparrow 2.6}$ | $66.1_{\uparrow 1.6}$ |
| + SFT | $41.9_{\uparrow 6.1}$ | $46.2_{\uparrow 8.8}$ | $30.3_{\uparrow 6.3}$ | $58.7_{\uparrow 1.6}$ | $69.5_{\uparrow 5.0}$ |

# 4 AUTOCODEINSTRUCT

**AutoCodeInstruct** Besides the evaluation benchmarks, we further construct AutoCodeInstruct, a training dataset of comparable quality to AutoCodeBench. Specifically, we collect data generated during the AutoCodeGen process that does not overlap with AutoCodeBench, and repeatedly sample `DeepSeek-V3-0324`. Problems with excessively high pass rates (>80%) or low pass rates (<40%) are filtered out to ensure both solvability and appropriate difficulty. We further apply a two-stage deduplication strategy (MinHash + LLM-as-Judge) across existing code benchmarks. The resulting dataset contains 37K verifiable problems spanning 20 programming languages.

**Training Setup** We conduct RL experiments via a two-stage GRPO (Shao et al., 2024) training strategy to unleash the potential of AutoCodeInstruct, based on the `Qwen2.5-Coder-7B/32B-Instruct` models. Concretely, we apply a data filtering strategy by sampling the responses 15 times from the Instruct models and filter out easy problems with pass rates above 0.6. The remaining problems are divided into solve-partial and solve-none parts depending on whether the pass rate is zero. In the first stage, only the solve-partial problems join training, with a rollout size of 8. In the second stage, we incorporate both solve-partial and solve-none problems, and increase the rollout size to 16 to enable better exploration for harder problems. Besides, for all solve-partial and solve-none problems, we additionally obtain correct code solutions from `DeepSeek-V3-0324` and perform SFT on `Qwen2.5-Coder-7B/32B-Instruct`. Details of the training configurations are presented in Appendix I.

**Results** As shown in Table 6, after the first-stage GRPO, both models achieve noticeable gains on in-domain benchmarks (ACB-Full and ACB-Lite), which suggests that they begin to learn how to stably consolidate existing knowledge. The second-stage GRPO enables the models to tackle harder problems, effectively pushing their multilingual capability boundaries and leading to significant performance improvements. Surprisingly, the models also show consistent gains on out-of-domain multilingual benchmarks such as FullStackBench and McEval. The performance of `Qwen2.5-Coder-32B-Instruct` on the programming contest task LiveCodeBench-V6 (20250201–20250501) also improves by 4 points. In addition, after the SFT stage, both models

achieve even larger performance improvements than those obtained from GRPO, thanks to the distilled correct code solutions from `DeepSeek-V3-0324`. These results indicate that AutoCodeInstruct enhances the comprehensive code generation capability of models and demonstrate the effectiveness of our approach in synthesizing high-quality training datasets for code LLMs.

## 5 RELATED WORK

**Code Generation Benchmarks** The rapid evolution of code LLMs, ranging from open-source models (Roziere et al., 2023; Zhu et al., 2024; Hui et al., 2024b) to proprietary LLMs (Anthropic, 2025a; OpenAI, 2024; 2025a; Gemini, 2025) series, has reshaped code generation, create a demand for robust and contemporary code generation benchmarks. Pioneering benchmarks like HumanEval (Chen et al., 2021) and MBPP (Austin et al., 2021) established foundational correctness on small Python tasks but suffer from contamination and limited language coverage. Later benchmarks target more complex settings, such as competition-level challenges (Hendrycks et al., 2021; Li et al., 2022; Jain et al., 2025; Wang et al., 2025b; Zheng et al., 2025) and multilingual scenarios (Cassano et al., 2022; Peng et al., 2024; Zhang et al., 2024; Jimenez et al., 2024; Chai et al., 2025; Bytedance, 2025; Zhang et al., 2025b;a;b). McEval (Chai et al., 2025) is a massively multilingual benchmark covering 40 languages for generation, explanation, and completion tasks. FullStackBench (Bytedance, 2025) assesses LLMs in realistic, multi-domain scenarios across 16 languages, employing a novel execution environment. However, due to the challenges of manual annotation, these benchmarks suffer from issues such as limited diversity and insufficient difficulty, making them difficult to scale in line with the evolving demand for high-quality evaluation. By comparison, our AutoCodeBench series adopts a fully automated and scalable approach to create realistic, diverse, and high-difficulty tasks. A recent trend, exemplified by the SWE-Bench series (Jimenez et al., 2024; Zan et al., 2025; Rashid et al., 2025; He et al., 2025), focuses on evaluating LLMs in real-world software engineering tasks such as GitHub issue solving, thereby assessing models' comprehensive capabilities beyond atomic-level code generation. By comparison, AutoCodeBench specifically targets LLMs' atomic-level code generation abilities, which remain a crucial foundation for overall model performance.

**Code Data Synthesis** To reduce dependence on manually curated data, a growing body of research explores automatic data synthesis to augment the training of Code LLMs (Luo et al., 2024; Wei et al., 2024b; Zheng et al., 2024; Wu et al., 2024; Yu et al., 2024; Ahmad et al., 2025; Xu et al., 2025). For instance, Evol-Instruct (Luo et al., 2024) uses heuristic prompts to guide LLMs in evolving existing programming problems, thereby increasing their diversity and difficulty. OSS-Instruct (Wei et al., 2024b) prompts LLMs to generate new coding problems and solutions from raw, open-source code snippets. KodCode (Xu et al., 2025) synthesizes a broad spectrum of Python coding tasks—including questions, solutions, and test cases—and ensures correctness through a systematic self-verification procedure. Some other methods focus on model self-improvement (Wu et al., 2024; Wei et al., 2024a; Chen et al., 2025b; Zhou et al., 2025; Zhang et al., 2025c). For instance, Inverse-Instruct (Wu et al., 2024) is a self-improvement technique that generates new instructions by "back-translating" code from an LLM's own training set, reducing the need to distill from more powerful proprietary models. Collectively, these data synthesis methods significantly reduce the reliance on manual curation and enable the continuous expansion of the problem space for training. Our work extends this paradigm of automation from data augmentation to the benchmark creation process. By leveraging extensive LLM-sandbox interaction, our pipeline not only automates the synthesis of verifiable test problems but can also be naturally repurposed for synthesizing high-quality training datasets.

## 6 FUTURE WORKS

Ensuring high-quality and reliable code data synthesis remains fundamentally challenging. Although the sandbox provides strong guarantees regarding the correctness of code solutions and their alignment with test functions, the intrinsic quality of the synthesized programming problems and the completeness of their test coverage cannot be fully ensured. AutoCodeGen incorporates an LLM-as-Critic stage and careful prompt engineering to mitigate such risks, yet these mechanisms inevitably operate under uncertainty. To further improve reliability, future iterations of AutoCodeGen will incorporate repeated verification rounds using increasingly capable LLMs. Double- or triple-check validation loops are expected to improve accuracy, albeit at the cost of increased computational overhead and reduced pipeline efficiency.

Moreover, repository-level evaluation and data synthesis represent a more realistic and demanding setting, while the current multi-logic analysis provides only an initial bridge between fine-grained function behavior and higher-level software engineering reasoning. Moving forward, we aim to extend AutoCodeGen toward SWE-Bench– and Terminal-Bench–style domains, enabling automated synthesis of repo-level tasks together with the stateful sandbox environments required for such scenarios. This represents a crucial step toward scalable and fully autonomous code evaluation frameworks.

## 7 CONCLUSION

In this paper, we explored the large-scale and automated construction of code generation benchmarks. We introduce AutoCodeGen, an automated workflow based on LLM-Sandbox interaction, designed to generate multilingual verifiable code data without any manual annotation. Through this novel approach, we have successfully built AutoCodeBench, a large-scale, human-free code generation benchmark. AutoCodeBench contains 3,920 problems, evenly distributed across 20 programming languages, and is characterized by its high difficulty, practicality, and diversity. We also provide AutoCodeBench-Lite and AutoCodeBench-Complete, for efficient and high-quality evaluation of both chat and base LLMs. Our evaluation of more than 40 open-source and proprietary LLMs reveals that even the most advanced models still face challenges when confronted with the complex and diverse multilingual tasks set by AutoCodeBench. Besides, we construct AutoCodeInstruct, a large-scale, high-quality multilingual training dataset, and validate its effectiveness through GRPO.

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

CONTENTS OF APPENDIX

Table 7: Comparison of Accuracy, Upper Bound, and Model Performance.

| | Average | Python | C++ | Java | JS | Go | Shell |
|---|---|---|---|---|---|---|---|
| Problem Accuracy | 87.6 | 83.5 | 88.0 | 86.0 | 89.0 | 86.0 | 93.3 |
| Current Upper Bound | $66.9_{\triangle 20.7}$ | $61.7_{\triangle 21.8}$ | $71.5_{\triangle 16.5}$ | $76.1_{\triangle 9.9}$ | $58.2_{\triangle 30.8}$ | $65.4_{\triangle 20.6}$ | $68.6_{\triangle 24.7}$ |
| Claude Opus 4 (Reasoning) | $44.6_{\triangle 43.0}$ | $40.3_{\triangle 43.2}$ | $44.1_{\triangle 43.9}$ | $55.9_{\triangle 30.1}$ | $38.6_{\triangle 50.4}$ | $37.2_{\triangle 48.8}$ | $51.6_{\triangle 41.7}$ |

## A  MANUAL VERIFICATION

Although our pipeline enforces quality control through specifications and an LLM-as-Critic mechanism, we further validate AutoCodeBench with human annotators. We employ a Human-LLM collaboration approach for data quality validation. Specifically, we design prompts in the native languages of the annotators and use the `DeepSeek-R1-0528` to generate detailed reasoning processes and checklist-based annotation results. The prompt is shown in Figure 10. During the annotation process, we assume that the programming problems are completely correct. The primary task of the annotators is to assess the correctness of the test functions and their alignment with the programming problem, based on the LLM's output. We allow for test cases that may not cover all boundary conditions, focusing primarily on the correctness of the test functions rather than their comprehensiveness. The annotators pay particular attention to the following aspects:

- Whether the function names, class names, variable definitions, and return types are consistent with the problem description;
- Whether the test cases exhibit randomness or non-reproducibility;
- Whether the test cases contradict the logic presented in the problem statement;
- Whether there are any precision issues with the test cases;
- Whether the test functions include test cases that are not addressed in the problem description.

We calculate the problem accuracy rates for different programming languages (Python, C++, Java, JavaScript, Go, Shell), as shown in Table 7. The results indicate that, despite the presence of some noisy data, our benchmark model still demonstrates high accuracy (87.6%). Furthermore, even after removing the noise, the current SOTA model shows significant room for improvement ($\triangle 43.0$), further validating the high difficulty level of our benchmark. The performance of Claude Opus 4 (Reasoning) in these six languages are only 44.6($\triangle 43.0$), highlighting the significant potential for improvement. Besides, we find that, compared to logic errors in the problem description and errors in the test functions, the most frequently occurring issue is **incomplete problem descriptions**. For example, some test functions reference class or function names that are essential but not explicitly mentioned in the problem description, or they require natural language outputs for edge cases that are not explicitly specified in the problem statement, leading to mismatches between the generated code and the test functions. Interestingly, we observe similar issues in manually annotated benchmarks, highlighting the significant challenge of creating comprehensive and accurate programming problems for annotators.

## B  SETUP OF BENCHMARK COMPARISONS

### B.1  MULTI-LOGIC

AutoCodeBench contains tasks that demand executing multiple functionalities, such as implementing both an addition and a multiplication function simultaneously. In contrast, the tasks in other benchmarks are almost exclusively focused on implementing a single core functionality.

### B.2  DIFFICULTY

We rate the difficulty of each benchmark based on the performance of `DeepSeek-V3-0324`. Specifically, benchmarks with pass@1 below 40 are assigned five stars; those between 40–50 receive four stars (e.g., LiveCodeBench-v6: 46.9, AutoCodeBench: 48.1); between 50–60 receive three

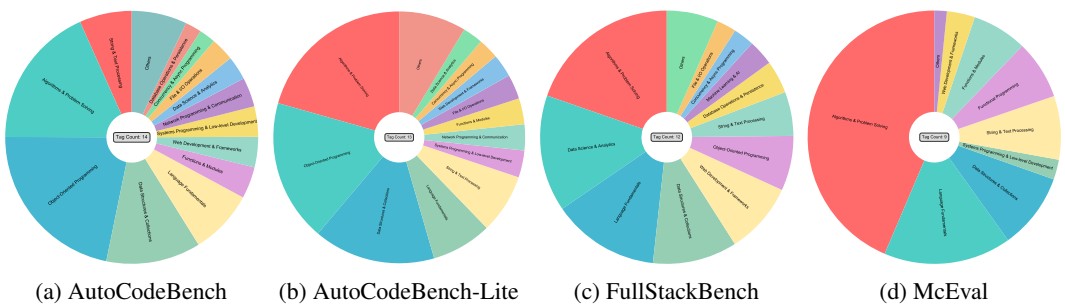

Figure 5: Category Distribution of Different Benchmarks.

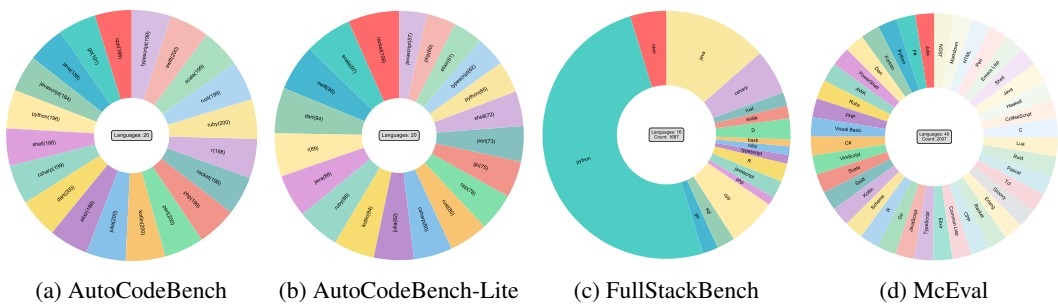

Figure 6: Language Distribution of Different Benchmarks.

stars; between 60–80 receive two stars (e.g., FullStackBench: 67.0, McEval: 72.3); and those above 80 receive one star. HumanEval and MBPP are excluded from evaluation due to extensive data leakage and its overly simple problems. For reference, `DeepSeek-V3 (October 2024)` already achieves a pass@1 of 91.5 in HumanEval.

### B.3 CATEGORY DISTRIBUTION AND LANGUAGE DISTRIBUTION

We prompt `Claude Sonnet 4` to generate 20 language-agnostic category labels for classification:

- **Core Programming Concepts**: *Language Fundamentals*, *Functions & Modules*, *Object-Oriented Programming*, *Functional Programming*, *Memory Management & Performance*, *Error Handling & Debugging*

- **Data and Algorithms**: *Data Structures & Collections*, *Algorithms & Problem Solving*, *String & Text Processing*, *File & I/O Operations*, *Concurrency & Async Programming*

- **Application Domains**: *Network Programming & Communication*, *Database Operations & Persistence*, *Web Development & Frameworks*, *Mobile & Cross-platform Development*, *Systems Programming & Low-level Development*

- **Advanced Topics and Tooling**: *Data Science & Analytics*, *Machine Learning & AI*, *Testing & Quality Assurance*, *Development Tools & Ecosystem*

In addition to AutoCodeBench, we conduct task tagging and language distribution analysis for AutoCodeBench-Lite, FullStackBench, and McEval. The results are presented in Figures 5 and 6. FullStackBench demonstrates comparable category diversity to AutoCodeBench(-Lite) but suffers from an imbalanced language distribution. In contrast, McEval exhibits a well-balanced multilingual distribution but lacks diversity and balance in its category coverage. Our AutoCodeBench(-Lite) achieves the most comprehensive category coverage while maintaining a balanced multilingual distribution, enabling thorough and accurate evaluation of LLMs' multilingual code generation capabilities.

Table 8: Pass@1 (%) performance of different base models for 3-shot AutoCodeBench-Complete.

| Count | Average | Python 50 | Cpp 50 | Java 50 | JS 50 | Go 50 | Shell 50 | Csharp 50 | Dart 50 | Elixir 50 | Julia 50 | Kotlin 50 | Perl 50 | PHP 50 | Racket 50 | R 50 | Ruby 50 | Rust 50 | Scala 50 | Swift 50 | TS 50 |
|---|---|---|---|---|---|---|---|---|---|---|---|---|---|---|---|---|---|---|---|---|---|
| **30B+ Models** | | | | | | | | | | | | | | | | | | | | | |
| DeepSeek-Coder-V2-Base | **39.0** | 24.0 | 32.0 | 40.0 | 44.0 | 34.0 | 26.0 | 64.0 | 38.0 | 52.0 | 46.0 | 56.0 | 38.0 | 36.0 | 32.0 | 26.0 | 40.0 | 26.0 | 36.0 | 42.0 | 48.0 |
| Qwen2.5-Coder-32B | 35.5 | 36.0 | 34.0 | 32.0 | 32.0 | 38.0 | 34.0 | 58.0 | 30.0 | 42.0 | 38.0 | 52.0 | 40.0 | 32.0 | 30.0 | 26.0 | 30.0 | 18.0 | 30.0 | 34.0 | 44.0 |
| Qwen2.5-72B | 35.9 | 32.0 | 22.0 | 38.0 | 40.0 | 22.0 | 34.0 | 62.0 | 22.0 | 42.0 | 38.0 | 46.0 | 42.0 | 46.0 | 28.0 | 26.0 | 38.0 | 28.0 | 28.0 | 30.0 | 54.0 |
| **~8B Models** | | | | | | | | | | | | | | | | | | | | | |
| Seed-Coder-8B-Base | **31.6** | 26.0 | 22.0 | 40.0 | 30.0 | 32.0 | 12.0 | 54.0 | 24.0 | 48.0 | 30.0 | 48.0 | 28.0 | 36.0 | 22.0 | 26.0 | 32.0 | 18.0 | 20.0 | 36.0 | 48.0 |
| OpenCoder-8B-Base | 26.1 | 22.0 | 6.0 | 28.0 | 34.0 | 30.0 | 24.0 | 52.0 | 10.0 | 42.0 | 32.0 | 28.0 | 26.0 | 24.0 | 20.0 | 20.0 | 28.0 | 14.0 | 26.0 | 14.0 | 42.0 |
| Qwen2.5-Coder-7B | 24.6 | 20.0 | 10.0 | 22.0 | 28.0 | 24.0 | 14.0 | 46.0 | 8.0 | 46.0 | 32.0 | 42.0 | 30.0 | 30.0 | 14.0 | 20.0 | 18.0 | 16.0 | 24.0 | 14.0 | 34.0 |
| DeepSeek-Coder-6.7B-Base | 22.9 | 20.0 | 14.0 | 26.0 | 34.0 | 18.0 | 18.0 | 50.0 | 8.0 | 44.0 | 20.0 | 38.0 | 28.0 | 18.0 | 12.0 | 14.0 | 34.0 | 6.0 | 10.0 | 4.0 | 42.0 |
| Qwen3-8B-Base | 22.6 | 20.0 | 14.0 | 18.0 | 34.0 | 20.0 | 12.0 | 50.0 | 6.0 | 34.0 | 26.0 | 24.0 | 32.0 | 30.0 | 8.0 | 20.0 | 30.0 | 8.0 | 14.0 | 16.0 | 36.0 |

Table 9: Pass@1 (%) performance of different models for AutoCodeBench-Lite.

| | Average | Python | Cpp | Java | JS | Go | Shell | Csharp | Dart | Elixir | Julia | Kotlin | Perl | PHP | Racket | R | Ruby | Rust | Scala | Swift | TS |
|---|---|---|---|---|---|---|---|---|---|---|---|---|---|---|---|---|---|---|---|---|---|
| **Count** | | 65 | 78 | 88 | 57 | 75 | 72 | 80 | 94 | 61 | 82 | 84 | 73 | 60 | 106 | 89 | 88 | 80 | 97 | 95 | 62 |
| *Current Upper Bound* | *100.0* | *100.0* | *100.0* | *100.0* | *100.0* | *100.0* | *100.0* | *100.0* | *100.0* | *100.0* | *100.0* | *100.0* | *100.0* | *100.0* | *100.0* | *100.0* | *100.0* | *100.0* | *100.0* | *100.0* | *100.0* |
| **Proprietary Models and 200B+ Open-source Models** | | | | | | | | | | | | | | | | | | | | | |
| Claude Opus 4.1 (20250805) | 69.9 | 61.5 | 62.8 | 65.9 | 68.4 | 68.0 | 70.8 | 90.0 | 69.1 | 78.7 | 73.2 | 73.8 | 67.1 | 55.0 | 80.2 | 68.5 | 72.7 | 67.5 | 56.7 | 67.4 | 77.4 |
| Claude Sonnet 4 (20250514) | 62.0 | 53.9 | 60.3 | 60.2 | 50.9 | 64.0 | 65.3 | 78.8 | 61.7 | 63.9 | 53.7 | 66.7 | 64.4 | 61.7 | 76.4 | 59.6 | 62.5 | 57.5 | 56.7 | 52.6 | 66.1 |
| Claude Opus 4.1 (20250805) | 63.8 | 53.8 | 61.5 | 61.4 | 63.2 | 61.3 | 62.5 | 83.7 | 62.8 | 63.9 | 68.3 | 73.8 | 61.6 | 48.3 | 68.9 | 60.7 | 61.4 | 65.0 | 56.7 | 60.0 | 72.6 |
| Claude Sonnet 4 (20250514) | 59.8 | 44.6 | 61.5 | 60.2 | 57.9 | 56.0 | 65.3 | 83.8 | 56.4 | 62.3 | 53.7 | 69.1 | 65.8 | 55.0 | 71.7 | 50.6 | 56.8 | 56.3 | 49.5 | 49.5 | 71.0 |
| GPT-5 (20250807) | 67.0 | 66.2 | 64.1 | 52.3 | 73.7 | 66.7 | 73.6 | 80.0 | 67.0 | 72.1 | 61.0 | 64.3 | 62.5 | 65.0 | 61.9 | 71.9 | 70.5 | 68.7 | 58.8 | 68.4 | 80.6 |
| o3-high (20250416) | 63.2 | 61.5 | 59.0 | 60.2 | 64.9 | 33.3 | 68.1 | 71.3 | 67.0 | 63.9 | 62.2 | 70.2 | 64.4 | 55.0 | 51.9 | 60.7 | 69.3 | 72.5 | 57.7 | 74.7 | 77.4 |
| o4-mini (2025-04-16) | 60.5 | 63.1 | 55.1 | 54.6 | 57.9 | 45.3 | 62.5 | 66.3 | 66.0 | 73.8 | 56.1 | 67.9 | 64.4 | 53.3 | 46.2 | 50.6 | 70.5 | 68.8 | 58.8 | 63.2 | 72.6 |
| GPT4.1 (2025-04-14) | 56.9 | 49.2 | 57.7 | 48.9 | 52.6 | 38.7 | 47.2 | 83.8 | 54.3 | 59.0 | 58.5 | 70.2 | 61.6 | 55.0 | 53.8 | 47.2 | 58.0 | 63.8 | 48.5 | 55.8 | 79.0 |
| Grok-4 | 63.0 | 60.9 | 64.1 | 55.7 | 56.1 | 61.3 | 68.1 | 82.5 | 55.3 | 59.0 | 63.4 | 69.1 | 63.0 | 41.7 | 72.6 | 59.6 | 73.9 | 65.0 | 60.8 | 54.7 | 66.1 |
| Gemini2.5 Pro | 59.1 | 56.9 | 59.7 | 59.1 | 54.4 | 50.7 | 62.5 | 75.0 | 64.9 | 62.3 | 64.6 | 71.4 | 58.9 | 51.7 | 53.8 | 60.7 | 63.6 | 37.5 | 53.6 | 57.9 | 61.3 |
| Gemini2.5 Flash | 52.9 | 49.2 | 56.4 | 54.6 | 50.9 | 45.3 | 45.8 | 75.0 | 56.4 | 45.0 | 51.2 | 63.1 | 54.8 | 40.0 | 34.0 | 59.6 | 58.0 | 58.8 | 46.4 | 51.6 | 61.3 |
| DeepSeek-V3.1 | 57.5 | 53.8 | 61.5 | 60.2 | 52.6 | 37.3 | 47.2 | 80.0 | 57.4 | 67.2 | 61.0 | 63.1 | 60.3 | 50.0 | 55.7 | 57.3 | 60.2 | 61.2 | 48.5 | 55.8 | 58.1 |
| DeepSeek-V3.1 | 52.2 | 44.6 | 51.3 | 64.8 | 43.9 | 33.3 | 43.1 | 72.5 | 43.6 | 59.0 | 56.1 | 57.1 | 61.6 | 50.0 | 51.9 | 55.1 | 51.1 | 50.0 | 42.3 | 48.4 | 66.1 |
| DeepSeek-Coder-V2-Instruct | 40.5 | 29.2 | 37.2 | 37.5 | 26.3 | 42.7 | 36.1 | 66.3 | 30.9 | 52.5 | 34.2 | 52.4 | 48.0 | 40.0 | 40.6 | 40.5 | 47.7 | 30.0 | 34.0 | 39.0 | 46.8 |
| Hunyuan-TurboS | 50.3 | 43.1 | 46.2 | 54.6 | 45.6 | 46.7 | 43.1 | 75.0 | 43.6 | 68.9 | 56.1 | 56.0 | 53.4 | 48.3 | 42.5 | 50.6 | 52.3 | 42.5 | 39.2 | 43.2 | 64.5 |
| GLM-4.5-enable | 55.0 | 56.1 | 54.7 | 56.1 | 49.0 | 54.2 | 59.4 | 76.3 | 39.5 | 59.3 | 52.4 | 63.1 | 61.6 | 50.0 | 49.5 | 57.8 | 58.0 | 50.7 | 42.9 | 52.1 | 61.3 |
| Kimi-K2-0905-preview | 53.7 | 46.2 | 44.9 | 51.1 | 52.6 | 49.3 | 51.4 | 73.7 | 58.5 | 65.6 | 51.2 | 56.0 | 56.2 | 50.0 | 60.4 | 48.3 | 48.9 | 45.0 | 48.5 | 51.6 | 66.1 |
| ERNIE-X1-Turbo-32K | 44.4 | 50.8 | 26.3 | 37.9 | 49.1 | 48.0 | 32.4 | 46.8 | 28.7 | 53.5 | 63.0 | 42.9 | 53.4 | 38.3 | 45.3 | 49.4 | 50.0 | 31.2 | 42.7 | 45.3 | 61.3 |
| Qwen3-235B-A22B-Thinking-2507 | 57.3 | 55.4 | 52.6 | 54.6 | 64.9 | 53.3 | 62.5 | 80.0 | 50.0 | 68.9 | 56.1 | 47.6 | 50.7 | 46.7 | 65.1 | 60.7 | 62.5 | 57.5 | 44.3 | 51.6 | 66.1 |
| Qwen3-Coder-480B-A35B-Instruct | 51.5 | 52.3 | 44.9 | 59.1 | 45.6 | 42.7 | 52.8 | 68.8 | 37.2 | 59.0 | 56.1 | 56.0 | 57.5 | 46.7 | 57.6 | 43.8 | 53.4 | 38.8 | 40.2 | 53.7 | 67.7 |
| Qwen3-235B-A22B-Instruct-2507 | 49.8 | 43.1 | 47.4 | 55.7 | 43.9 | 44.0 | 54.2 | 73.8 | 38.3 | 45.9 | 46.3 | 50.0 | 67.1 | 43.3 | 53.8 | 51.7 | 47.7 | 40.0 | 37.1 | 48.4 | 66.1 |
| Seed1.6-Thinking-250715 | 53.9 | 56.9 | 60.3 | 58.0 | 49.1 | 58.7 | 48.6 | 73.8 | 41.5 | 52.5 | 59.8 | 60.7 | 54.8 | 43.3 | 51.9 | 55.1 | 60.2 | 46.3 | 40.2 | 47.4 | 62.9 |
| Seed1.6-enabled (250615) | 53.2 | 52.3 | 57.7 | 51.1 | 45.6 | 60.0 | 55.6 | 62.5 | 43.6 | 52.5 | 57.3 | 54.8 | 57.5 | 41.7 | 52.8 | 60.7 | 58.0 | 43.8 | 41.2 | 51.6 | 66.1 |
| Seed1.6-disabled (250615) | 48.8 | 41.5 | 51.3 | 50.0 | 47.4 | 50.7 | 43.1 | 78.8 | 42.6 | 52.5 | 50.0 | 52.4 | 56.2 | 33.3 | 50.0 | 46.1 | 47.7 | 48.8 | 29.9 | 40.0 | 71.0 |
| **200B Open-source Models** | | | | | | | | | | | | | | | | | | | | | |
| GLM-4.5-Air-enable | 46.2 | 53.9 | 44.9 | 45.5 | 45.6 | 50.7 | 40.3 | 72.5 | 38.3 | 50.8 | 42.7 | 44.1 | 56.2 | 45.0 | 41.5 | 48.3 | 47.7 | 33.8 | 29.9 | 40.0 | 67.7 |
| Qwen3-Next-80B-A3B-Thinking | 46.3 | 50.8 | 41.0 | 48.9 | 59.6 | 42.7 | 45.8 | 76.2 | 24.5 | 42.6 | 47.6 | 39.3 | 60.3 | 41.7 | 29.2 | 53.9 | 54.5 | 38.7 | 38.1 | 44.7 | 61.3 |
| Qwen3-Next-80B-A3B-Instruct | 42.6 | 43.1 | 35.9 | 46.6 | 42.1 | 34.7 | 47.2 | 66.2 | 35.1 | 44.3 | 43.9 | 28.6 | 58.9 | 45.0 | 36.8 | 43.8 | 40.9 | 38.7 | 30.9 | 42.1 | 59.7 |
| Qwen3-32B | **47.6** | 50.8 | 43.6 | 46.6 | 43.9 | 46.8 | 48.6 | 72.5 | 36.2 | 55.7 | 47.6 | 44.1 | 60.3 | 45.0 | 35.9 | 52.8 | 44.3 | 43.8 | 37.1 | 46.3 | 64.5 |
| Qwen3-14B | 40.7 | 46.2 | 39.7 | 39.8 | 42.1 | 37.3 | 40.3 | 62.5 | 24.5 | 52.5 | 41.5 | 33.3 | 50.7 | 33.3 | 41.5 | 41.6 | 43.2 | 42.5 | 22.7 | 33.7 | 61.3 |
| Qwen3-8B | 28.9 | 29.2 | 24.4 | 19.3 | 35.1 | 38.7 | 29.2 | 51.3 | 18.1 | 39.3 | 28.1 | 23.8 | 45.2 | 25.0 | 8.5 | 34.8 | 37.5 | 18.8 | 18.6 | 20.0 | 56.5 |
| Qwen3-1.7B | 10.8 | 21.5 | 2.6 | 4.6 | 14.0 | 10.7 | 15.3 | 18.8 | 6.4 | 13.1 | 9.8 | 7.1 | 26.0 | 10.0 | 1.9 | 13.5 | 18.2 | 0.0 | 1.0 | 8.4 | 27.4 |
| Qwen3-32B | 32.3 | 29.2 | 25.6 | 31.8 | 31.6 | 32.0 | 36.1 | 56.3 | 25.5 | 36.1 | 31.7 | 33.3 | 43.8 | 38.3 | 25.5 | 22.5 | 35.2 | 25.0 | 23.7 | 25.3 | 51.6 |
| Qwen3-14B | 27.8 | 24.6 | 20.5 | 30.7 | 38.6 | 16.0 | 22.2 | 47.5 | 16.0 | 31.2 | 19.5 | 34.5 | 45.2 | 41.7 | 22.6 | 29.2 | 28.4 | 25.0 | 13.4 | 22.1 | 45.2 |
| Qwen3-8B | 21.4 | 26.2 | 6.4 | 31.8 | 28.1 | 20.0 | 20.8 | 38.8 | 16.0 | 24.6 | 17.1 | 23.8 | 30.1 | 23.3 | 18.9 | 15.7 | 25.0 | 8.8 | 11.3 | 11.6 | 43.6 |
| Qwen3-1.7B | 7.3 | 7.7 | 0.0 | 1.1 | 7.0 | 5.3 | 12.5 | 8.8 | 3.2 | 8.2 | 6.1 | 8.3 | 21.9 | 13.3 | 2.8 | 9.0 | 12.5 | 0.0 | 2.1 | 4.2 | 22.6 |
| Qwen2.5-Coder-32B-Instruct | 37.0 | 33.9 | 26.9 | 38.6 | 40.4 | 28.0 | 30.6 | 57.5 | 29.8 | 52.5 | 31.7 | 46.4 | 52.1 | 43.3 | 30.2 | 34.8 | 40.9 | 28.8 | 22.7 | 35.8 | 48.4 |
| Qwen2.5-Coder-7B-Instruct | 21.5 | 23.1 | 6.4 | 21.6 | 24.6 | 12.0 | 22.2 | 42.5 | 16.0 | 32.8 | 18.3 | 33.3 | 34.3 | 25.0 | 17.9 | 21.4 | 25.0 | 5.0 | 16.5 | 10.5 | 33.9 |
| Qwen2.5-Coder-1.5B-Instruct | 10.2 | 12.3 | 3.9 | 3.4 | 17.5 | 12.0 | 19.4 | 16.3 | 6.4 | 31.2 | 1.2 | 14.3 | 23.3 | 8.3 | 5.7 | 6.7 | 12.5 | 0.0 | 3.1 | 3.2 | 21.0 |
| DeepSeek-Coder-33B-Instruct | 27.7 | 23.1 | 25.6 | 23.9 | 35.1 | 37.3 | 26.4 | 52.5 | 14.9 | 59.0 | 11.0 | 46.4 | 39.7 | 33.3 | 23.6 | 14.6 | 27.3 | 6.3 | 26.8 | 13.7 | 33.9 |
| DeepSeek-Coder-6.7B-Instruct | 19.9 | 20.0 | 11.5 | 20.5 | 28.1 | 29.3 | 16.7 | 40.0 | 10.6 | 39.3 | 15.9 | 39.3 | 24.7 | 16.7 | 13.2 | 10.1 | 23.9 | 8.8 | 11.3 | 4.2 | 32.3 |
| Seed-Coder-8B-Instruct | 32.7 | 20.0 | 25.6 | 31.8 | 42.1 | 28.0 | 20.8 | 52.5 | 27.7 | 45.9 | 34.1 | 41.7 | 41.1 | 41.7 | 36.8 | 25.8 | 33.0 | 18.7 | 28.9 | 21.1 | 46.8 |
| OpenCoder-8B-Instruct | 20.1 | 13.9 | 11.5 | 20.5 | 17.5 | 25.3 | 23.6 | 30.0 | 17.0 | 32.8 | 12.2 | 22.6 | 30.1 | 23.3 | 15.1 | 15.7 | 29.6 | 7.5 | 16.5 | 14.7 | 30.7 |

## B.4 PROBLEM LENGTH

We use `Qwen2.5-32B-Instruct` tokenizer to calculate the problem length. For Multi-SWE-Bench, problem length is not reported, as its tasks depend on extremely large multi-file repositories.

## C THE RESULTS OF AUTOCODEBENCH-LITE

The pass@1 values of AutoCodeBench-Lite is shown in Table 9.

## D THE RESULTS OF AUTOCODEBENCH-COMPLETE

The pass@1 values of AutoCodeBench-Complete is shown in Table 8.

## E LEADERBOARDS

The leaderboards of ACB-Full and ACB-Lite are shown in Figure 7 and 8.

## F PERFORMANCE ANALYSIS OF SCALING LAWS

Figure 9 compares parameter scaling and test-time sampling scaling across different models. The parameter scaling law (left) shows significant variation between models, with `Qwen3 (Think)` Series demonstrating the steepest scaling curve, indicating that chain-of-thought reasoning particularly

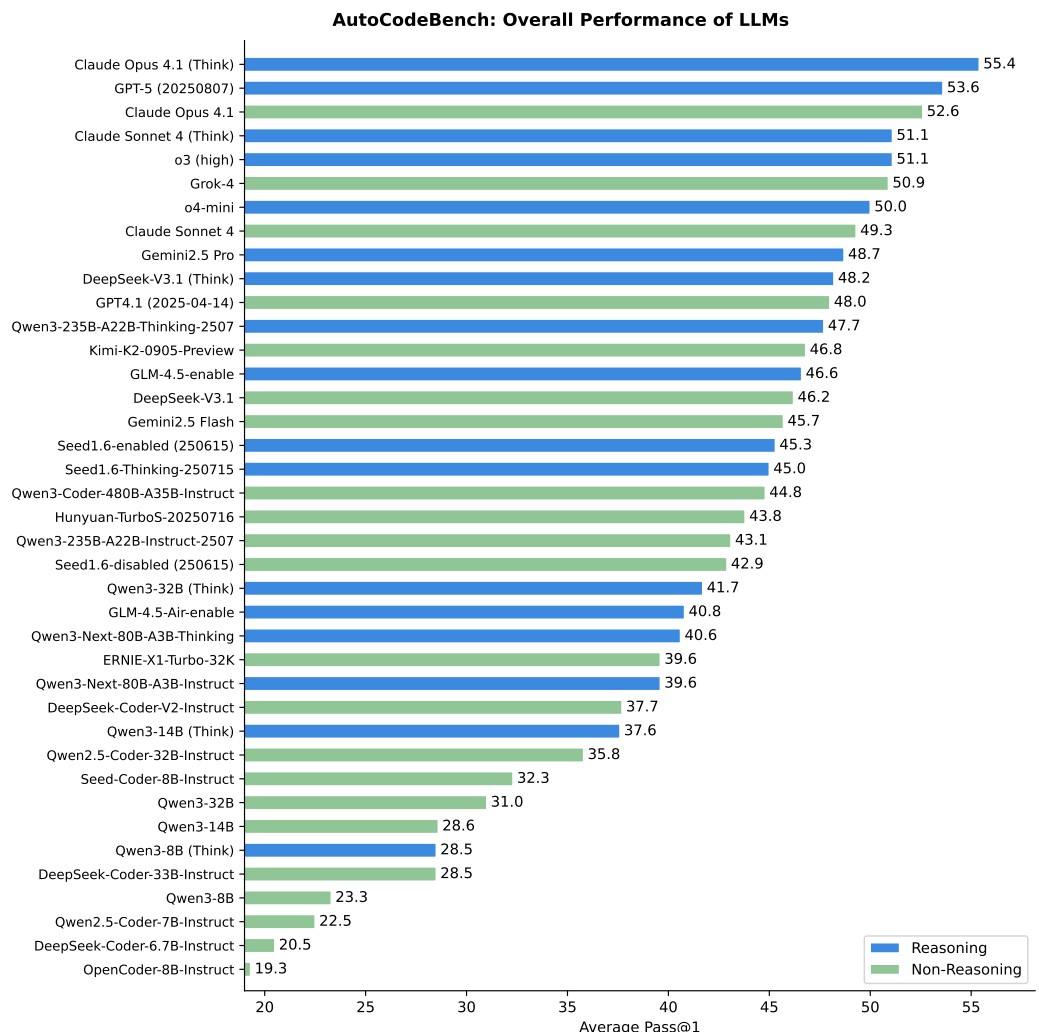

Figure 7: AutoCodeBench leaderboard showing Pass@1 performance of various LLMs.

benefits larger models. The test-time sampling scaling law (right) reveals more uniform behavior, with three models showing similar improvement rates from increased sampling during inference. These results suggest that while test-time sampling provides consistent benefits regardless of model size, reasoning capabilities scale more aggressively with model size.

## G    BASELINES

We evaluate a diverse set of open-source models with sizes ranging from 1.5B to 1T parameters, as well as leading proprietary models. These models are classified based on their families:

- **OpenAI**: `GPT-5` (OpenAI, 2025b), `o3` and `o4-mini` (OpenAI, 2025), and `GPT4.1` (OpenAI, 2025a).
- **Claude**: `Claude Opus 4.1` (Anthropic, 2025b) and `Claude Sonnet 4` (Anthropic, 2025a).
- **Gemini**: `Gemini 2.5 Pro` and `Gemini 2.5 Flash` (Gemini, 2025).
- **DeepSeek**: `DeepSeek-V3.1` (DeepSeek-AI, 2025a;b) and `DeepSeek-Coder` Series (DeepSeek-AI et al., 2024; Guo et al., 2024).

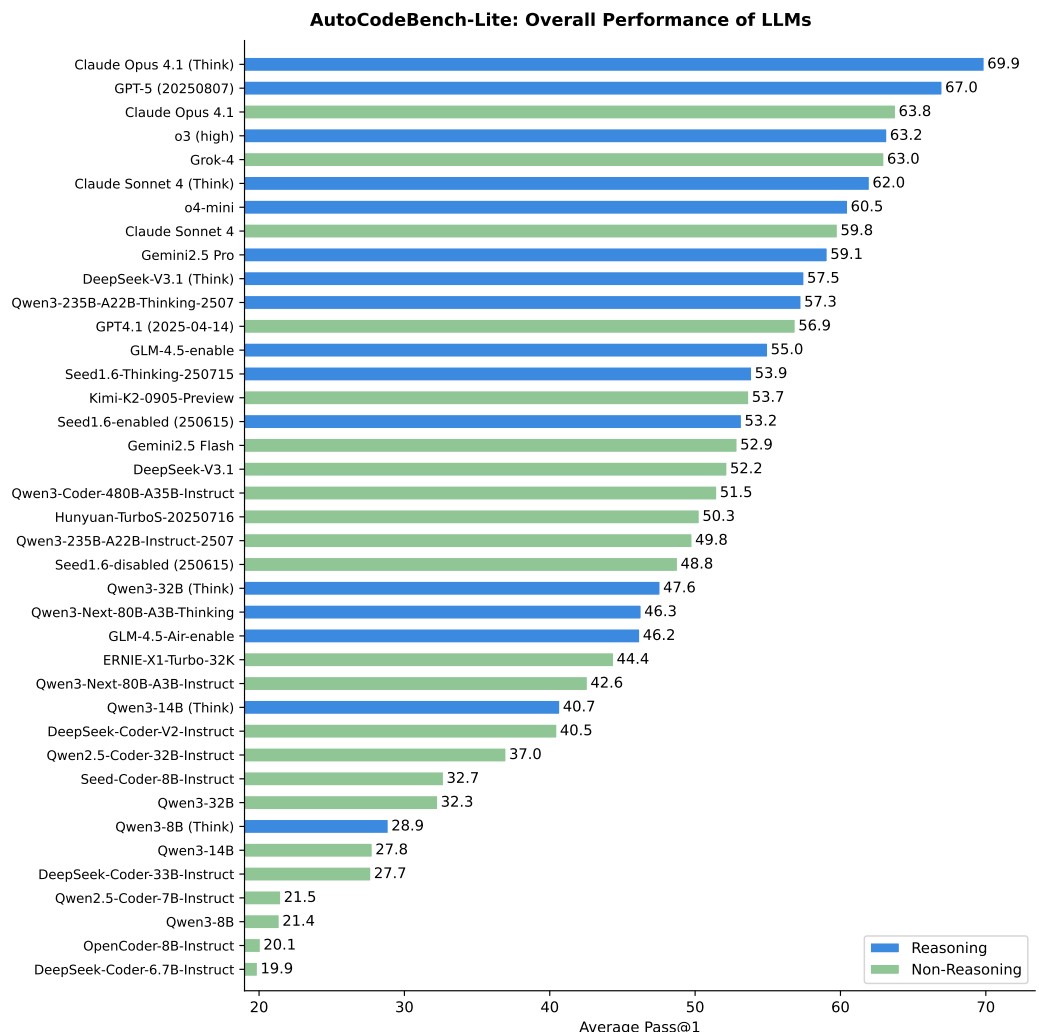

Figure 8: AutoCodeBench-Lite leaderboard showing Pass@1 performance of various LLMs.

- **Hunyuan**: `Hunyuan-TurboS` (Tencent, 2025).
- **Qwen**: `Qwen3-Next-80B-A3B` (Qwen, 2025), `Qwen3-235B-A22B-Thinking-2507`, `Qwen3-235B-A22B-Instruct-2507`, and `Qwen3` Series (Yang et al., 2025), `Qwen3-Coder-480B-A35B-Instruct` (Qwen, 2025), `Qwen2.5-Coder` Series (Hui et al., 2024a).
- **Seed**: `Seed1.6-Thinking` (Seed, 2025), `Seed1.6` (Seed, 2025) and `Seed-Coder-8B` (Seed et al., 2025).
- **GLM**: `GLM-4.5` and `GLM-4.5-Air` (Zhipu, 2025).
- **Other Models**: `ERNIE-X1-Turbo-32K` (Baidu, 2025), `Kimi-K2` (Kimi-Team, 2025), and `OpenCoder-8B` (Huang et al., 2025).

# H   HYPOTHESES ON MODEL BIAS IN THE GENERATION PROCESS

It is well-known that models exhibit inherent biases, particularly their tendency to favor their own outputs—a common phenomenon in automated data synthesis and evaluation tasks (Panickssery et al., 2024; Chen et al., 2025a). Our automated workflow is no exception to this issue. While completely eliminating such bias is challenging, we employ several mitigation strategies. Specifically,

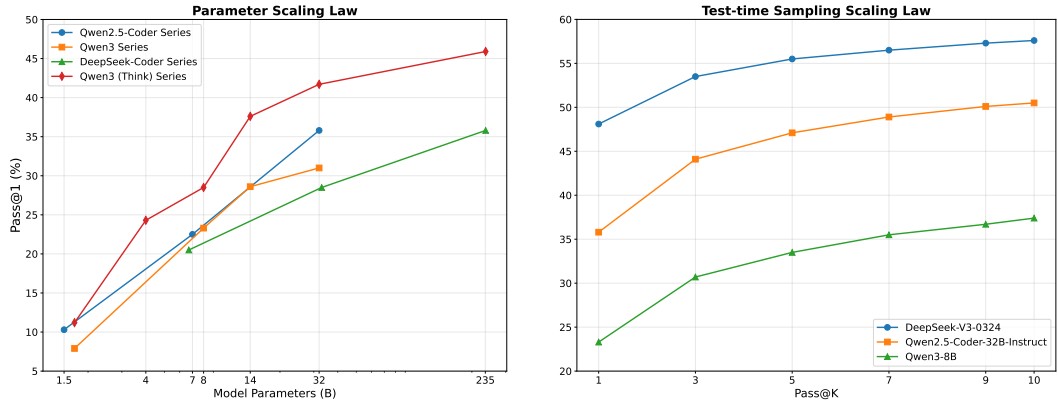

Figure 9: Scaling laws for different models.

Table 10: The average pass@1 scores and rankings of models at different stages.

|  | Initial Stage (Rank) | After Simple Problem Filtering (Rank) | After Critic Filtering (Rank) |
|---|---|---|---|
| DeepSeek-V3-0324 | 47.1 (3) | 25.7 $_{-21.4}$ (4) | 31.6 $_{+5.9}$ (4) |
| DeepSeek-R1-0528 | 48.9 (2) | 28.7 $_{-20.2}$ (2) | 36.2 $_{+7.5}$ (2) |
| o3 | 46.4 (4) | 28.1 $_{-18.3}$ (3) | 34.9 $_{+6.8}$ (3) |
| Gemini2.5 Pro | 51.4 (1) | 31.6 $_{-19.8}$ (1) | 38.7 $_{+7.0}$ (1) |
| Qwen2.5-Coder-32B-Instruct | 39.9 (5) | 17.1 $_{-22.8}$ (5) | 22.0 $_{+4.9}$ (5) |

we intentionally only use DeepSeek series models in the workflow to prevent bias from affecting other model families. We hypothesize that using `DeepSeek-V3-0324` for code generation and `DeepSeek-R1-0528` for the Critic process may introduce favorable bias toward DeepSeek families. To counteract this, we employ `DeepSeek-Coder-V2-Lite` during the simple problem filtering phase, creating a "push-and-pull" mechanism that balances potential biases across different stages.

To quantitatively assess bias, we sampled 3,600 data points across six programming languages (Python, C++, Java, JS, Go, and Shell) and tracked performance changes at each generation stage, as shown in Table 10. The results reveal nuanced bias patterns: simple problem filtering negatively impacts smaller models (`Qwen2.5-Coder-32B-Instruct`) more than DeepSeek series, while the Critic process benefits `DeepSeek-R1-0528` but surprisingly provides greater improvements to reasoning models (`o3` and `Gemini 2.5 Pro`) than to `DeepSeek-V3-0324`. This suggests that model bias depends not only on model family but also on factors like model size and reasoning modes. Furthermore, as mutual distillation between models from different families continues, this bias becomes increasingly difficult to measure. In conclusion, we believe that our automated process may introduce a favorable bias toward the DeepSeek family of models, but the impact is minimal.

# I   AUTOCODEINSTRUCT EXPERIMENTAL DETAILS

We apply separate data filtering for the two Instruct models, as the performance varies across models. This results in 8684 solve-partial ($0 < pass\_rate < 0.6$) prompts and 4882 solve-none ($pass\_rate = 0$) prompts for `Qwen2.5-Coder-7B-Instruct`, 10518 solve-partial and 3294 solve-none prompts for `Qwen2.5-Coder-32B-Instruct`. During the two-stage GRPO training, the batch size is set to 128 and 64, respectively for 32B and 7B experiments. The learning rate is set to $1 \times 10^{-6}$ and the maximum input/output lengths are 8192/8192. The 32B model is trained for 60 steps in the first stage and 80 steps in the second stage, while the 7B model is trained for 70 steps in both stages. During SFT, the batch size is set to 64, the gradient accumulation steps are set to 2, the learning rate is $1 \times 10^{-5}$, and both the 32B and 7B models are trained for two epochs. For evaluation, we adopt greedy decoding for all the models and the maximum output length is set to 16384.

```
#CONTEXT#
You are a **{language}** test development expert.
The system will provide you with two inputs:
1. **[QUESTION]** – The problem description, outlining the functionality and constraints that the code under test
should fulfill.
2. **[TEST_FUNCTION]** – The test script, containing several test cases.

Your task is to rigorously review the **[TEST_FUNCTION]** to ensure it truly verifies the requirements of the
**[QUESTION]**, and provide structured review results for the data annotators.

# OBJECTIVE #
Review the **[TEST_FUNCTION]** according to the following audit rules (in fixed order), providing a boolean value
and a 20-40 word justification for each:
1. Naming/Signature Mismatch – Function names, classes, variables, and return types in the test do not match the
ones described in the question.
2. Randomness/Non-Determinism – The test cases contain random factors, and the results are unstable (e.g., calling
random without a seed or relying on system time).
3. Incorrect Test Target – The test case verifies functionality that is not described in the problem.
4. Precision Handling Issues – High precision requirements use == instead of math.isclose() or similar approximate
comparisons, only using them when precision is needed.
5. Exception Swallowing – The test case catches exceptions (try-except) which obscure the actual errors.
6. Unexecutable Test – The default environment dependencies are okay, but the entry point (like if __name__ ==
"__main__":) is missing, making the test not executable.
7. Irrelevant Requirements – The test case checks for functionality not required by the problem (reasonable edge
cases are exceptions).
8. Other Issues – Any defects not covered by the above rules.

# STYLE #
- Structured, concise, engineering tone
- Standard JSON format; fields should use snake_case
- Justification should be 20-40 words, in Chinese

# TONE #
Professional, objective, direct

# AUDIENCE #
Data annotators with senior development experience, who need to judge the quality of test scripts based on this.

#RESPONSE#
Only output the following JSON structure (without Markdown code block tags):

```json
{
        "rule_results": [
        {
                "rule": "Naming/Signature Mismatch",
                "result": true,
                "reason": "Example: Function name 'add' does not match 'sum' in the problem"
        },
        // … 8 items in total in the fixed order
        ],
        "summary": {
                "overall_pass": false,
                "failed_rules": ["Naming/Signature Mismatch", "Randomness/Non-Determinism"]
                "key_points": "Random factors lead to unstable results; test target deviates from the problem's
requirement"
        }
}

- rule_results: List of 8 rules in fixed order (result should be true or false).
- summary.overall_pass: false if any rule result is false; otherwise true.
- summary.failed_rules: List of all failed rule descriptions; empty if all pass.
- summary.key_points: ≤60 words summarizing the main flaws.

# USER INPUT #
<QUESTION>
{question}
</QUESTION>
<TEST_FUNCTION>
{test_function}
</TEST_FUNCTION>
```

Figure 10: The English prompt of annotation and critic.

## J  MULTILINGUAL CODE SANDBOX SERVICE

This service offers a secure and high-performance environment for the compilation and execution of code in over 30 programming languages. It supports large-scale code data validation, making it suitable for high-volume, automated testing scenarios. Our multilingual sandbox has the following features:

- **Multilingual Support**: The service supports more than 30 programming languages, including popular ones like Python, JavaScript, Go, Java, C++, and Rust, providing versatility for various use cases.

- **Security Isolation**: Code execution is isolated within Docker containers, ensuring that each execution environment is separate. Additionally, iptables firewall rules are applied to maintain a high level of security, preventing unauthorized access or interference.

Table 11: Statistics of the 1,622 multi-logic programming problems.

| | #Problems | #Test Cases | #Langs | Prob Len | Solu Len | Difficulty (E/M/H) |
|---|---|---|---|---|---|---|
| MLPP | 1,622 | 16,131 | 20 | 576.4 | 610.0 | 238/315/1069 |

- **Smart Code Integration**: The system automatically manages the integration of function code with testing code. It adapts to language-specific syntax, ensuring seamless execution without requiring manual intervention for code merging.

- **High Performance**: Powered by a Gunicorn multi-process architecture, the sandbox supports concurrent execution of multiple code instances, making it capable of handling a high volume of requests efficiently.

- **RESTful API**: The service provides a clean and easy-to-use HTTP-based API, allowing developers to interact with the sandbox programmatically, whether for integrating into larger applications or automating tasks.

- **Extensive Language Support**: Beyond the mainstream languages, the sandbox also supports emerging and niche languages, allowing it to cater to a wide variety of development environments and user needs.

- **Custom Execution Environments**: Users can configure specific environments for their tasks, enabling tailored execution conditions based on their unique requirements.

## K    PROMPTS FOR AUTOMATED WORKFLOW

The prompt of generating code solution is shown in Figure 11.

The prompt of generating test function is shown in Figure 12.

The prompt of generating programming problem is shown in Figure 13.

The prompt of LLM-as-Critic is shown in Figure 10.

The prompt of translating languages is shown in Figure 14.

## L    MULTI-LOGIC TASK ANALYSIS

First, we provide a clear definition of multi-logic programming tasks:

**Multi-Logic Programming Problem (MLPP).**   MLPP is a programming task whose correct solution requires implementing and coordinating multiple core logical units—such as functions, classes, or modules. Each logical unit corresponds to an independent semantic responsibility or algorithmic objective.

We further divide Multi-Logic Problems into two categories:

- **Intra-Logic Problems**: multiple logical units collaborate to achieve a single overarching functional goal.

- **Inter-Logic Problems**: multiple logical units address distinct functional goals, each independently contributing to the full solution.

Following the structure of Table 3, we perform a detailed analysis on the 1,622 multi-logic problems in AutoCodeBench. As shown in Table 11, multi-logic problems exhibit substantially longer problem descriptions and longer canonical solutions compared to the overall dataset, reflecting richer instruction structure and greater intrinsic difficulty.

In addition, we used `DeepSeek-V3.2-Exp` to compute (1) the number of logical units contained in each multi-logic problem, and (2) the distribution of Inter-Logic and Intra-Logic types. On average, each problem contains **3.37** logical units. Among them, **389** are Inter-Logic, **1,223** are Intra-Logic,

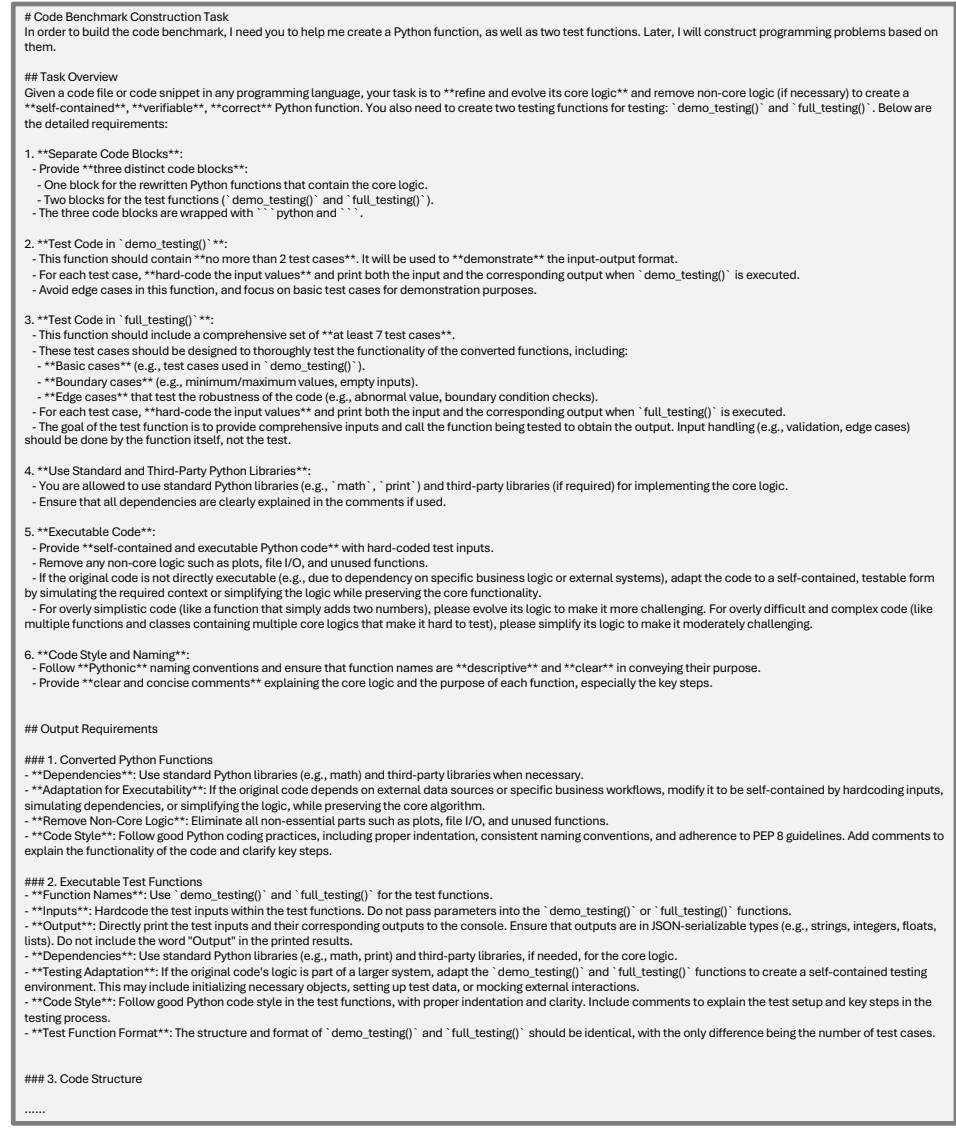

# Code Benchmark Construction Task
In order to build the code benchmark, I need you to help me create a Python function, as well as two test functions. Later, I will construct programming problems based on them.

## Task Overview
Given a code file or code snippet in any programming language, your task is to **refine and evolve its core logic** and remove non-core logic (if necessary) to create a **self-contained**, **verifiable**, **correct** Python function. You also need to create two testing functions for testing: `demo_testing()` and `full_testing()`. Below are the detailed requirements:

1. **Separate Code Blocks**:
   - Provide **three distinct code blocks**:
     - One block for the rewritten Python functions that contain the core logic.
     - Two blocks for the test functions (`demo_testing()` and `full_testing()`).
   - The three code blocks are wrapped with ```python and ```.

2. **Test Code in `demo_testing()`**:
   - This function should contain **no more than 2 test cases**. It will be used to **demonstrate** the input-output format.
   - For each test case, **hard-code the input values** and print both the input and the corresponding output when `demo_testing()` is executed.
   - Avoid edge cases in this function, and focus on basic test cases for demonstration purposes.

3. **Test Code in `full_testing()`**:
   - This function should include a comprehensive set of **at least 7 test cases**.
   - These test cases should be designed to thoroughly test the functionality of the converted functions, including:
     - **Basic cases** (e.g., test cases used in `demo_testing()`).
     - **Boundary cases** (e.g., minimum/maximum values, empty inputs).
     - **Edge cases** that test the robustness of the code (e.g., abnormal value, boundary condition checks).
   - For each test case, **hard-code the input values** and print both the input and the corresponding output when `full_testing()` is executed.
   - The goal of the test function is to provide comprehensive inputs and call the function being tested to obtain the output. Input handling (e.g., validation, edge cases) should be done by the function itself, not the test.

4. **Use Standard and Third-Party Python Libraries**:
   - You are allowed to use standard Python libraries (e.g., `math`, `print`) and third-party libraries (if required) for implementing the core logic.
   - Ensure that all dependencies are clearly explained in the comments if used.

5. **Executable Code**:
   - Provide **self-contained and executable Python code** with hard-coded test inputs.
   - Remove any non-core logic such as plots, file I/O, and unused functions.
   - If the original code is not directly executable (e.g., due to dependency on specific business logic or external systems), adapt the code to a self-contained, testable form by simulating the required context or simplifying the logic while preserving the core functionality.
   - For overly simplistic code (like a function that simply adds two numbers), please evolve its logic to make it more challenging. For overly difficult and complex code (like multiple functions and classes containing multiple core logics that make it hard to test), please simplify its logic to make it moderately challenging.

6. **Code Style and Naming**:
   - Follow **Pythonic** naming conventions and ensure that function names are **descriptive** and **clear** in conveying their purpose.
   - Provide **clear and concise comments** explaining the core logic and the purpose of each function, especially the key steps.

## Output Requirements

### 1. Converted Python Functions
- **Dependencies**: Use standard Python libraries (e.g., math) and third-party libraries when necessary.
- **Adaptation for Executability**: If the original code depends on external data sources or specific business workflows, modify it to be self-contained by hardcoding inputs, simulating dependencies, or simplifying the logic, while preserving the core algorithm.
- **Remove Non-Core Logic**: Eliminate all non-essential parts such as plots, file I/O, and unused functions.
- **Code Style**: Follow good Python coding practices, including proper indentation, consistent naming conventions, and adherence to PEP 8 guidelines. Add comments to explain the functionality of the code and clarify key steps.

### 2. Executable Test Functions
- **Function Names**: Use `demo_testing()` and `full_testing()` for the test functions.
- **Inputs**: Hardcode the test inputs within the test functions. Do not pass parameters into the `demo_testing()` or `full_testing()` functions.
- **Output**: Directly print the test inputs and their corresponding outputs to the console. Ensure that outputs are in JSON-serializable types (e.g., strings, integers, floats, lists). Do not include the word "Output" in the printed results.
- **Dependencies**: Use standard Python libraries (e.g., math, print) and third-party libraries, if needed, for the core logic.
- **Testing Adaptation**: If the original code's logic is part of a larger system, adapt the `demo_testing()` and `full_testing()` functions to create a self-contained testing environment. This may include initializing necessary objects, setting up test data, or mocking external interactions.
- **Code Style**: Follow good Python code style in the test functions, with proper indentation and clarity. Include comments to explain the test setup and key steps in the testing process.
- **Test Function Format**: The structure and format of `demo_testing()` and `full_testing()` should be identical, with the only difference being the number of test cases.

### 3. Code Structure
......

Figure 11: The prompt of generating code solution. Due to the excessive length of the prompt, we have omitted the latter part.

and **10** fall into both categories. Opus 4.1 achieved 50.6% and 51.5% performance on the Inter-Logic and Intra-Logic types, respectively.

Figure 16 provides an example of an Inter-Logic problem. The task requires implementing three separate classes, each serving a different functional purpose—clearly corresponding to independent logical goals. Figure 17 shows a more complex hybrid case. The problem requires implementing a Rational class containing multiple internal logical units that jointly support the overarching goal of "rational number representation and operations," making them Intra-Logic. At the same time, the auxiliary function "parse_rationals" handles a separate goal of "string parsing," forming an Inter-Logic relationship with the Rational class.

The results of Opus 4.1 and these cases indicate that, in multi-logic problems, more complex instructions, greater logical demands, and intricate relationships among logical units pose greater challenges for LLMs.

```
I'll provide you with a Python code, a test function call (including inputs) that uses this Python function, and the test output obtained after executing that test function call.

Please combine the provided inputs and outputs into an assert statement, and place these `assert` statements inside a new `test` function. You can only use the inputs and outputs provided by me. Please do not create your own or modify the test cases.

Please generate a Python test function using assert statements. You will be provided with:
- Python Code: The function(s) to be tested.
- Two Test Function Calls (including inputs): Python code demonstrating how the function is called with various inputs.
- Test Outputs: The results obtained after executing the provided test function calls.
You will receive two sets of test cases:

Important Considerations for assert statements:
- DO NOT create or modify any test cases. Use only the inputs and outputs provided.
- Avoid assertions that might differ due to floating-point precision across machines. If the original output involves floating-point numbers, and the problem context suggests
it, consider using math.isclose() or asserting within a reasonable tolerance if the test cases inherently involve such comparisons and precision is a concern. However,
prioritize direct equality == if the provided outputs are exact.
- All assert statements must be placed within a function named test().
- Please create **two separate** test functions for each of the two sets of test function calls I provide, ensuring they do not interfere with each other.
- Two test function names are both "def test()". They are placed in two code blocks.

Here is an example:
**Code**:
```python
from math import sqrt

def is_prime(n):
    if n < 2:
        return False
    for i in range(2, int(sqrt(n)) + 1):
        if n % i == 0:
            return False
    return True

def nth_prime(n):
    if n < 1:
        return None
    count = 1
    i = 2
    while count < n:
        i += 1
        if is_prime(i):
            count += 1
    return i
```

**Test Function Call 1**:
```python
def demo_testing():
    # Hard-coded test inputs
    test_cases = [1]

    for n in test_cases:
        # Print the input
        print(f"Input: {n}")

        # Core logic
        result = nth_prime(n)

        # Output (JSON-serializable)
        print(f"The {n}th prime number is: {result}")

if __name__ == "__main__":
    demo_testing()
```

**Test Case Results1**:
```
'Input: 1\nThe 1th prime number is: 2'
```

**Test Function Call 2**
......
```

Figure 12: The prompt of generating test function. Due to the excessive length of the prompt, we have omitted the latter part.

## M  LLM-BASED ERROR ANALYSIS

In this section, we present an error type analysis. As shown in Appendix A, a fully automated pipeline inevitably introduces a certain amount of noise. To distinguish noisy data from genuinely difficult tasks, we designed a checklist-based template to systematically categorize error types. The checklist is as follows:

- **Model Solution Errors**
    - **Misunderstanding Problem**: The model fails to correctly interpret the problem statement, leading to a fundamentally incorrect solution.
    - **Failure to Comprehend Complex Instructions**: The model fails to correctly interpret or decompose multi-step or nuanced requirements in the problem statement, resulting in conceptually incomplete or irrelevant solutions.

You are an experienced programming tutor, adept at crafting **clear, concise, and educational programming problems**. I will supply you with a **self-contained, executable, and correct Python code**, along with one or more test functions designed to verify its correctness. Your task is to **generate a programming problem that directly corresponds to the given Python code and its test cases**.

Here is the code and test functions:
[Python code]
<<>>
[Python code end]

[test function demo]
<<<demo_test>>>
[test function demo end]

[test function]
<<<full_test>>>
[test function end]

Please ensure the problem you generate adheres to the following critical requirements:
1. Language Specification: Explicitly state that solutions must be implemented in Python.
2. Problem Description: Describe the problem concisely and unambiguously using plain language. Avoid technical jargon, unnecessary details, or solution hints.
3. Function/Class Naming: Only mention the exact function or class names used in the test functions. Do not include implementation-specific details beyond what's in the tests.
4. Input/Output Format: Define the input format (types, structure, value ranges). Define the expected output format. If necessary, specify some constraints (e.g., input size limits, allowed data types).
5. Example Usage: Use the test case in the aforementioned 'test function demo' to construct example usage. The number of test cases in this is usually no more than three. Do not modify or explain the test cases—just copy them verbatim.
6. No Solution Hints: The problem description must not reveal any code solution and any test cases beyond what's in the provided examples.

Please enclose the generated programming problem within <question> and </question> tags.

Figure 13: The prompt of generating programming problem.

Table 12: Top-10 error types.

| Type | Count |
| --- | --- |
| Unreasonable Test Cases | 313 |
| Algorithmic Logic Error | 218 |
| Missing Functionality | 157 |
| Misunderstanding Problem | 151 |
| Algorithmic Logic Error + Misunderstanding Problem | 121 |
| Insufficient Fundamental Knowledge | 108 |
| Algorithmic Logic Error + Missing Functionality | 106 |
| Algorithmic Logic Error + Unreasonable Test Cases | 73 |
| Algorithmic Logic Error + Insufficient Fundamental Knowledge | 57 |
| Omission of Edge Cases | 55 |

- **Omission of Edge Cases**: The model fails to handle extreme or special input conditions.
- **Missing Functionality**: The model understands the task but fails to implement all required components or subtasks in the generated code.
- **Algorithmic Logic Error**: The algorithmic approach or control logic contains mistakes leading to incorrect results.
- **Insufficient Fundamental Knowledge**: The model demonstrates a lack of understanding of basic programming syntax, data structures, or language semantics.
- **Problem Design Errors**
  - **Logical Contradiction in the Problem**: The problem statement contains internal inconsistencies or mutually exclusive constraints that make a correct solution impossible.
  - **Ambiguous Problem Statement**: The problem description is unclear or lacks explicit constraints, leading to multiple valid interpretations.
- **Test Design Errors**
  - **Unreasonable Test Cases**: The test data deviate from the problem definition.

We applied this checklist to analyze all 1,745 tasks that `Claude Opus 4.1` failed to solve, using `DeepSeek-V3.2-Exp` as the judge LLM. If an error was attributed to **Problem Design** or **Test Design**, the task was classified as low-quality. Otherwise, we considered it a valid high-difficulty task. The Top-10 error types are shown in Table 12.

In total, 538 tasks contained *Problem Design Errors* or *Test Design Errors*, accounting for 13.7% of all tasks. The remaining 1,207 tasks were judged to be high-quality and genuinely challenging.

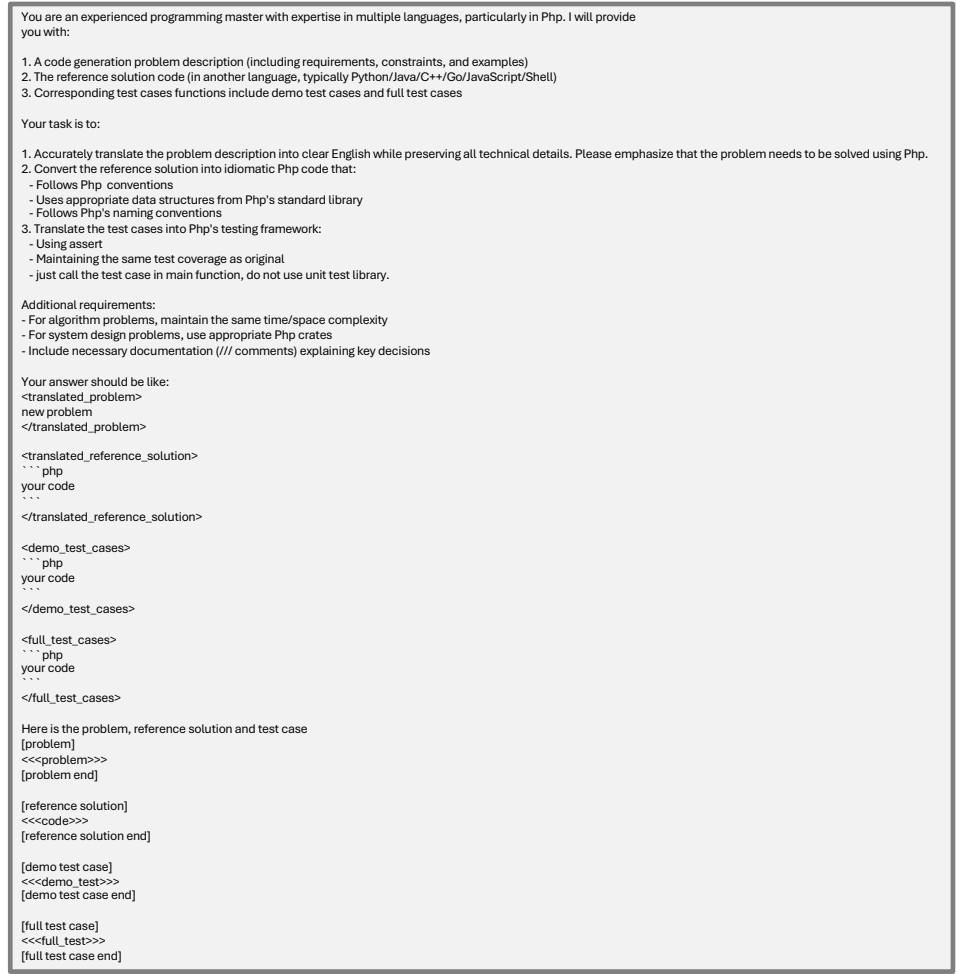

You are an experienced programming master with expertise in multiple languages, particularly in Php. I will provide you with:

1. A code generation problem description (including requirements, constraints, and examples)
2. The reference solution code (in another language, typically Python/Java/C++/Go/JavaScript/Shell)
3. Corresponding test cases functions include demo test cases and full test cases

Your task is to:

1. Accurately translate the problem description into clear English while preserving all technical details. Please emphasize that the problem needs to be solved using Php.
2. Convert the reference solution into idiomatic Php code that:
   - Follows Php conventions
   - Uses appropriate data structures from Php's standard library
   - Follows Php's naming conventions
3. Translate the test cases into Php's testing framework:
   - Using assert
   - Maintaining the same test coverage as original
   - just call the test case in main function, do not use unit test library.

Additional requirements:
- For algorithm problems, maintain the same time/space complexity
- For system design problems, use appropriate Php crates
- Include necessary documentation (/// comments) explaining key decisions

Your answer should be like:
<translated_problem>
new problem
</translated_problem>

<translated_reference_solution>
```php
your code
```
</translated_reference_solution>

<demo_test_cases>
```php
your code
```
</demo_test_cases>

<full_test_cases>
```php
your code
```
</full_test_cases>

Here is the problem, reference solution and test case
[problem]
<<<problem>>>
[problem end]

[reference solution]
<<>>
[reference solution end]

[demo test case]
<<<demo_test>>>
[demo test case end]

[full test case]
<<<full_test>>>
[full test case end]

Figure 14: The prompt of translating languages.

Among these, the dominant failure modes were *Algorithmic Logic Errors* and *Missing Functionality*. Figures 18 and 19 illustrate representative cases. After excluding low-quality tasks, `Opus 4.1` still achieves only 64.3%, demonstrating that the remaining tasks remain substantially challenging.

## N  THE ORDER OF I/O FORMAT

In this section, we elaborate on the design choice of generating test inputs prior to producing the programming problem description in the AutoCodeGen pipeline. Although human annotators typically define the input–output (I/O) format before writing tests, our automated workflow adopts the reverse order. We show that this alternative workflow is both feasible for LLMs and beneficial in practice. There are two valid sequences for synthesizing test cases and problem descriptions:

- Generate the test inputs first, followed by the programming problem (including its I/O specification);
- Generate the I/O specification first, followed by the test inputs.

Both workflows can be handled by modern LLMs. However, the first approach offers an additional advantage: the public test cases produced during this stage can be directly embedded into the final problem description, improving clarity and reducing ambiguity in the specification. Below is an example that illustrates why generating test inputs first is a reasonable design choice.

Consider a code solution such as:

```
Please implement the following three network-related functions in Go:

1. ReverseIPAddress: Accepts an IPv4 address of type net.IP and returns its reversed string representation (e.g.,
"192.168.1.1" becomes "1.1.168.192"). If the input is invalid, return an empty string.

2. ReputationLookup: Accepts an IPv4 address as a string and returns its reputation value as a string (e.g., "100"). If the
input is invalid, return an error.

3. HostsFromCIDR: Accepts a CIDR-formatted string (e.g., "192.168.1.0/24") and returns a string slice of all available host
IPs within that range (excluding the network and broadcast addresses). If the input is invalid, return an error.

Input/Output Requirements:
- All IP addresses must be in IPv4 format.
- Invalid inputs include malformed or unparsable addresses.
- The CIDR range must be valid and contain at least one available host address.

Example Usage:

// Test case 1: Reverse IP address
ip := net.ParseIP("192.168.1.1")
reversed := ReverseIPAddress(ip)
// reversed should be "1.1.168.192"

// Test case 2: Query reputation
ipStr := "8.8.8.8"
rep, err := ReputationLookup(ipStr)
// rep should be "100", err should be nil

// Test case 3: Get host list from CIDR
cidr := "192.168.1.0/29"
hosts, err := HostsFromCIDR(cidr)
// hosts should be ["192.168.1.1", "192.168.1.2", "192.168.1.3", "192.168.1.4", "192.168.1.5", "192.168.1.6"], err should
be nil
```

Figure 15: The prompt of generating code solution. Due to the excessive length of the prompt, we have omitted the latter part.

```
def cal_two_sum(a, b):
    return a + b
```

Using explicit specifications and few-shot examples, the model may generate a test-input function such as:

```
def private_test_input():
    test_cases = [
        [1, 2],
        [0, 0], # boundary test
        [9999999999999999, 3333333333] # stress test
    ]
```

When constructing the programming problem, the model infers the input domain from the generated test cases and produces a description such as: *Write a function named cal_two_sum that computes the sum of two non-negative integers.* If instead the model generates test cases containing floating-point values:

```
def private_test_input():
    test_cases = [
        [1, 2],
        [0, 0],
        [4.243, 4.222], # stress test
        [-0.45, 888], # boundary test
    ]
```

The model naturally adapts the problem description accordingly: *Write a function named cal_two_sum that computes the sum of two floating-point numbers.* Therefore, When test inputs are generated first, the model derives the I/O format by analyzing the input domain represented in the test cases. When the I/O format is specified first, the model constructs matching test cases based on the declared constraints. Both approaches are logically consistent, but generating test inputs first leverages the model's ability to generalize from concrete examples and simplifies the subsequent problem construction stage.

Finally, regardless of the workflow, AutoCodeGen employs an LLM-as-Critic verification stage to ensure strict alignment between the generated programming problem and the test function. Any inconsistent, ambiguous, or underspecified cases are filtered out during this procedure. This ensures the final benchmark maintains high quality and internal coherence.

```
# 3D Camera System Implementation

## Problem Description
You are tasked with implementing a 3D camera system for a game engine. The system should include:
1. A `Vector` class for 3D mathematical operations
2. An `AABB` (Axis-Aligned Bounding Box) class for collision detection
3. A `Camera` class that represents a view frustum in 3D space

The camera should be able to:
- Track its position and orientation in 3D space
- Calculate its view frustum's bounding box
- Determine if objects (represented as AABBs) are within its view frustum

## Class Requirements

### Vector Class
```cpp
class Vector {
public:
    float x, y, z;

    Vector(float x=0, float y=0, float z=0);
    Vector operator+(const Vector& other) const;
    Vector operator-(const Vector& other) const;
    Vector operator*(float scalar) const;
    float dot(const Vector& other) const;
    Vector cross(const Vector& other) const;
    float magnitude() const;
    Vector normalized() const;
    void print() const;
};
```

### AABB Class
```cpp
class AABB {
public:
    Vector min, max;

    AABB(Vector min=Vector(), Vector max=Vector());
    bool contains(const Vector& point) const;
    void print() const;
};
```

### Camera Class
```cpp
class Camera {
public:
    Camera(const string& name="DefaultCamera",
           const Vector& pos=Vector(),
           const Vector& rotator=Vector(),
           float fov=60.0f,
           float zFar=1000.0f,
           float zNear=0.1f,
           float width=800.0f,
           float height=600.0f);

…
};
```

## Example Usage
```cpp
…
```

## Problem Specifications
…

## Constraints
…

## Notes
…
```

Figure 16: Example of an inter-logic problem. The full content is partially omitted due to length.

```
# Rational Number Operations

Implement a class `Rational` to represent and manipulate rational numbers (fractions) with the following capabilities:

## Class Requirements

### Private Members
- `long long numerator`: The numerator of the fraction
- `long long denominator`: The denominator of the fraction
- `long long gcd(long long a, long long b) const`: A helper function to compute the greatest common divisor of two numbers
- `void normalize()`: A helper function to reduce the fraction to its simplest form and ensure the denominator is positive

### Public Members
#### Constructor
- `Rational(long long num = 0, long long denom = 1)`: Creates a Rational number with given numerator and denominator
(defaults to 0/1). Throws `invalid_argument` if denominator is zero.

#### Arithmetic Operators
- `Rational operator+(const Rational& other) const`: Adds two Rational numbers
- `Rational operator-(const Rational& other) const`: Subtracts two Rational numbers
- `Rational operator*(const Rational& other) const`: Multiplies two Rational numbers
- `Rational operator/(const Rational& other) const`: Divides two Rational numbers (throws `domain_error` if dividing by
zero)

#### Comparison Operators
- `bool operator==(const Rational& other) const`: Checks equality of two Rational numbers
- `bool operator<(const Rational& other) const`: Checks if this Rational is less than another
- `bool operator>(const Rational& other) const`: Checks if this Rational is greater than another

#### Conversion Functions
- `double to_double() const`: Converts the Rational number to a double
- `string to_string(bool show_parens = false) const`: Returns a string representation of the Rational number. When
`show_parens` is true, negative numbers should be enclosed in parentheses. The string should be in mixed number form when
appropriate (e.g., "2 1/3" for 7/3).

#### Getters
- `long long get_numerator() const`: Returns the numerator
- `long long get_denominator() const`: Returns the denominator

### Helper Function
- `vector<Rational> parse_rationals(const string& input)`: Parses a space-separated string of rational numbers (either in
"a/b" form or whole numbers) and returns a vector of Rational numbers.

## Example Usage

```cpp
Rational a(1, 2);
Rational b(1, 3);
Rational sum = a + b;
cout << sum.to_string(); // Outputs "5/6"

Rational c(7, 3);
cout << c.to_string(); // Outputs "2 1/3"

Rational d(-1, 4);
cout << d.to_string(true); // Outputs "(-1/4)"

string input = "1/2 3/4 -1/3 5";
vector<Rational> numbers = parse_rationals(input);
for (const auto& num : numbers) {
    cout << num.to_string() << " ";
}
// Outputs: "1/2 3/4 -1/3 5 "
```

## Problem Specifications
…
## Constraints
…

## Notes
…

Your solution will be tested against various cases including arithmetic operations, comparisons, and string conversions.
```

Figure 17: Example of a problem exhibiting both inter-logic and intra-logic components. The full content is partially omitted due to length.

```
[programming problem]
Write a JavaScript function `octalToBinary` that converts a given octal number string into its corresponding binary string.

Input requirements:
- The input is a string representing an octal number.
- The string may be empty or contain non-octal digit characters (characters other than 0-7).
- Valid octal numbers may include leading zeros (e.g., "0000").

Output requirements:
- If the input is a valid octal string, return the corresponding binary string representation.
- If the input is an empty string, return "Error: Empty input".
- If the input contains non-octal digit characters, return "Error: Invalid octal string".
- The binary output should not omit leading zeros (e.g., octal "7" should return "111" and not "111").

Example usage:
```javascript
const assert = require('assert');

const demoTesting = () => {
    assert.strictEqual(octalToBinary("123"), "1010011");
    assert.strictEqual(octalToBinary("7"), "111");
};
```
```

**[Private Test Function]**

```
const assert = require('assert');

const fullTesting = () => {
    assert.strictEqual(octalToBinary("123"), "1010011");
    assert.strictEqual(octalToBinary("7"), "111");
    assert.strictEqual(octalToBinary("0"), "0");
    assert.strictEqual(octalToBinary("777"), "111111111");
    assert.strictEqual(octalToBinary("100"), "1000000");
    assert.strictEqual(octalToBinary("12345670"), "1010011100101110111000");
    assert.strictEqual(octalToBinary(""), "Error: Empty input");
    assert.strictEqual(octalToBinary("abc"), "Error: Invalid octal string");
    assert.strictEqual(octalToBinary("89"), "Error: Invalid octal string");
    assert.strictEqual(octalToBinary("0000"), "0000000000");
};
```

**[Code Solution from Opus 4.1]**

```javascript
function octalToBinary(octalStr) {
    // Check for empty input
    if (octalStr === '') {
        return "Error: Empty input";
    }

    // Check if all characters are valid octal digits (0-7)
    for (let char of octalStr) {
        if (char < '0' || char > '7') {
            return "Error: Invalid octal string";
        }
    }

    // Convert each octal digit to 3-bit binary representation
    let binaryStr = '';
    for (let digit of octalStr) {
        const octalDigit = parseInt(digit);
        const binaryDigit = octalDigit.toString(2).padStart(3, '0');
        binaryStr += binaryDigit;
    }

    // Remove leading zeros, but keep at least one digit          Logic Error!
    while (binaryStr.length > 1 && binaryStr[0] === '0') {
        binaryStr = binaryStr.substring(1);
    }

    return binaryStr;
}
```

Figure 18: A case of Algorithmic logic errors.

```
[programming problem]
Write a Go function to find the lowest common ancestor of two nodes in a binary tree.

Given the root of a binary tree and two nodes p and q in the tree, implement the function `lowestCommonAncestor` to return
the lowest common ancestor (LCA) of these two nodes. The lowest common ancestor is defined as the lowest node that has both
p and q as descendants (where a node can be a descendant of itself).

The function signature is as follows:
```go
func lowestCommonAncestor(root, p, q *TreeNode) *TreeNode
```

The `TreeNode` is defined as:
```go
type TreeNode struct {
    Val   int
    Left  *TreeNode
    Right *TreeNode
}
```

Input constraints:
- All node values in the tree are unique.
- Both p and q exist in the given binary tree.
- The number of nodes in the tree ranges from [2, 10^5].

Example usage:
```go
import "testing"

func TestDemo(t *testing.T) {
    // Create the tree structure for test case 1
    node7 := &TreeNode{7, nil, nil}
    node4 := &TreeNode{4, nil, nil}
    node2 := &TreeNode{2, node7, node4}
    node6 := &TreeNode{6, nil, nil}
    ……
```

Notes:
- Ensure your solution handles various cases, including but not limited to: p and q in different subtrees, p being an
ancestor of q, q being an ancestor of p, p and q being the same node, etc.
- Do not modify the tree structure.
-  The time complexity should be as efficient as possible.
[Code Solution from Opus 4.1]
```go
func lowestCommonAncestor(root, p, q *TreeNode) *TreeNode {
    // Base case: if root is nil or root is one of the target nodes
    if root == nil || root == p || root == q {
        return root
    }

    // Recursively search for p and q in left and right subtrees
    left := lowestCommonAncestor(root.Left, p, q)
    right := lowestCommonAncestor(root.Right, p, q)

    // If both left and right are non-nil, current node is the LCA
    // This means p and q are in different subtrees
    if left != nil && right != nil {
        return root
    }

    // If only one subtree contains both nodes, return that result
    // If left is non-nil, both nodes are in left subtree
    if left != nil {
        return left
    }
    // Otherwise, both nodes are in right subtree (or both are nil)
    return right
}
```
```

*Missing the definition of TreeNode!*

Figure 19: A case of Missing functionality.