# OpenReview forum: "AutoCodeBench: Large Language Models are Automatic Code Benchmark Generators"
_ICLR.cc/2026/Conference — ICLR 2026 Poster_

### Official Review · Reviewer_5r4r · 2025-10-27

**Soundness:** 2
**Presentation:** 3
**Contribution:** 2
**Rating:** 4
**Confidence:** 5

**Summary:**

This paper introduces AutoCodeGen, a fully automated pipeline for generating multilingual code generation benchmarks without manual annotation. The core of the proposed method is generates test inputs using LLM, which are then executed in a secure sandbox to obtain ground-truth outputs. These verified input-output pairs are subsequently used to formulate a complete programming problem. Using this pipeline, the authors present AutoCodeBench, a large-scale benchmark comprising 3,920 problems across 20 programming languages, designed to be both difficult and diverse. The authors report that the benchmark reveals significant weaknesses in current state-of-the-art models, particularly on low-resource languages and multi-logic tasks. In addition, they release a companion instruction-tuning dataset (AutoCodeInstruct) to support future research.

**Strengths:**

1. The cost and scalability limitations of manual annotation are well-known, and automating this process is a valuable research direction.
2. The public release of the benchmark, a large-scale training dataset, and a high-concurrency multilingual sandbox are significant assets that could benefit the code generation research community.

**Weaknesses:**

1. While the engineering effort is commendable, the core concept of using LLMs for data synthesis in the coding domain is not new and builds upon a line of existing work (e.g., Evol-Instruct, OSS-Instruct).
2. A significant concern is the methodology used to generate problems for 14 of the 20 languages via "approximate language translation (sec. 2.1.5)". However, code translation is not a solved problem, and often fails to capture language-specific idioms, standard library conventions, or type system nuances. This approach raises questions about the quality and naturalness of the generated problems in these low-resource languages. It is unclear what the quality gap is between these "translated" problems and problems that could have been generated "natively" using the full pipeline. The paper would be strengthened by an analysis quantifying this gap.

**Questions:**

1. The benchmark was generated using a model from the DeepSeek family. To what extent are the benchmark's characteristics and the resulting model performance metrics biased by this choice of generator? It would be insightful to conduct a small-scale experiment where the key pipeline steps are run using a completely different model family (e.g., Llama 3, Claude 3, or Qwen). A comparison of the resulting problems (e.g., types, styles) and model performance on this new subset would help quantify the potential for generator bias. Meanwhile, I agree Line 1064.
2. The concept of "multi-logic" problems is interesting and presented as a key contribution. However, its definition in the paper feels somewhat informal. Could the authors provide a more rigorous definition? Furthermore, a deeper analysis of what makes these problems challenging would be valuable. Is the difficulty primarily a function of longer context and following multiple discrete instructions, or does it require a more fundamental type of compositional reasoning that current LLMs lack?

---

> ### Author Response · Authors · 2025-11-25
> **Response for W1**
>
> ## Weakness1: Novelty
> As we discuss in Section 5 (Related Work), code data synthesis is a rapidly growing and highly dynamic research area, with numerous influential works emerging each year.
>
> The works you mentioned (Evol-Instruct[1] and OSS-Instruct[2]) are indeed classic early efforts in this direction. However, **they do not involve test case generation**, and therefore cannot support RL training or RFT-style iterative refinement. Compared with more recent approaches that produce *verifiable* code data, such as KodCode[3] and OpenCodeInstruct[4], our AutoCodeGen pipeline introduces several key innovations:
>
> ### **(1) A novel three-stage test case generation paradigm**
>
> KodCode and OpenCodeInstruct directly ask LLMs to generate full test functions, which is often inefficient and tends to produce low-difficulty tests. In contrast, Our *input generation → sandbox execution → input–output integration* pipeline simultaneously improves efficiency, correctness, and scalability.
>
> ### **(2) Reverse problem generation**
>
> We first construct executable solutions and test functions, and only then generate the programming problem. This reverse workflow ensures that public test cases are embedded accurately and consistently, something prior pipelines do not support.
>
> ### **(3) Multilingual scalability**
>
> Existing verifiable synthesis methods primarily focus on Python. In contrast, AutoCodeGen extends to 20 programming languages, enabled by a multilingual sandbox and an approximate translation workflow that preserves diversity and coverage.
>
> ### **(4) Benchmark-oriented filtering**
>
> AutoCodeGen applies difficulty control, LLM-as-critic verification, and diversity sampling, producing AutoCodeBench. The resulting dataset is a balanced, challenging, and comprehensive benchmark, rather than raw training data.
>
> In summary, AutoCodeGen distinguishes itself through verifiability, multilingual coverage, efficient test case construction, reverse problem generation, and benchmark-quality filtering, which are not present in prior methods.
>
>
>
> > [1] WizardLM: Empowering large pre-trained language models to follow complex instructions, ICLR24
> >
> > [2] Magicoder: Empowering Code Generation with OSS-Instruct, ICML24
> >
> > [3] KodCode: A Diverse, Challenging, and Verifiable Synthetic Dataset for Coding, ACL25
> >
> > [4] OpenCodeInstruct: A Large-scale Instruction Tuning Dataset for Code LLMs, Arxiv25
>
> ##

---

> > ### Author Response · Authors · 2025-11-25
> > **Response for W2**
> >
> > ## Weakness2: Approximate language translation
> > We fully agree with the reviewer that code translation remains an unsolved problem, especially when translating into low-resource programming languages, and this inevitably introduces lossiness. Nevertheless, we chose to adopt approximate code translation for several reasons:
> >
> > - **Verifiability:** Although translated code may suffer from quality degradation, our sandbox-based execution provides a reliable safety net to ensure correctness.
> > - **Scalability:** Code translation provides a one-to-many pathway, making it substantially easier to scale the pipeline to a wide range of programming languages.
> > - **Data diversity:** Popular languages (e.g., Python) contain far richer and more diverse seed distributions than low-resource languages. Even with some translation loss, this approach allows us to obtain *larger and more diverse* datasets for low-resource languages.
> >
> > ### Small-Scale Experiment on Translation Quality
> >
> > Following the reviewer’s suggestion, we conduct a controlled comparison using 1,000 Python seeds and 1,000 Swift seeds. For Python seeds, we generate `<question, code solution, test func>` tuples and then translate them into Swift. For Swift seeds, we use the same pipeline to directly generate the tuples natively in Swift. All experiments were performed using DeepSeek-V3.2-Exp.
> >
> > The stepwise retention rates were:
> >
> > | Stage                                | python→swift | swift       |
> > | :----------------------------------- | :----------- | :---------- |
> > | Initial seeds                        | 1000 (100%)  | 1000 (100%) |
> > | Solution generation using LLMs       | 994          | 1000        |
> > | Sandbox execution                    | 935          | 834         |
> > | Test func generation using LLMs      | 930          | 825         |
> > | Sandbox execution                    | 894          | 750 (75.0%) |
> > | Code translation + sandbox execution | 475 (47.5%)  | —           |
> >
> > Thus, **Python→Swift exhibits a 47.5% retention rate**, highlighting the difficulty of cross-language translation. We then analyze the 419 translation failures using DeepSeek-V3.2-Exp-think. We found that **over 65%** are due to basic syntactic issues or incorrect test cases, with additional failures attributable to incorrect error-handling patterns—all consistent with the reviewer’s point that code translation remains a challenging and imperfect process.
> >
> > We then examine Critic pass rates using LLM-as-Critic:
> >
> > - Translated data passed **93/475 (20.7%)**
> > - Native Swift data passed **120/750 (16%)**
> >
> > Interestingly, despite translation difficulty, the *post-filter* acceptance rate of translated samples is higher.
> >
> > Finally, we designed a five-aspect checklist prompt to compare the “Swifty-ness” of the accepted translated samples (93) and native samples (120).
> >
> > | Aspect / score                      | python→swift | swift    |
> > | :---------------------------------- | :----------- | :------- |
> > | Naming & Style                      | 3.7          | 3.8      |
> > | Stdlib & API usage                  | 2.8          | 2.9      |
> > | Control-flow & idioms               | 3.4          | 3.4      |
> > | Optionals & Error Handling          | 3.6          | 3.5      |
> > | Naturalness & translation artifacts | 3.2          | 3.4      |
> > | **Total**                           | **16.7**     | **17.0** |
> >
> > The scores are extremely close, indicating that after sandbox validation and multi-stage filtering, translated data maintains a level of naturalness and language conformity comparable to native generation.
> >
> > These results collectively suggest that although approximate code translation is imperfect, **the AutoCodeGen pipeline’s verification, execution-based filtering, and LLM-as-Critic mechanisms effectively remove low-quality translations**.

---

> > > ### Author Response · Authors · 2025-11-25
> > > **Response for Q1 [1/2]**
> > >
> > > ## Q1: Running AutoCodeGen using different model
> > >
> > > **Experimental Setup**. We conduct a small-scale experiment using models from the Qwen family to further investigate the potential generator bias. Specifically, we used Qwen-Plus-250718 for solution generation, test generation, and question generation; Qwen2.5-14B-Instruct for simple problem filtering; and Qwen-Plus -250718-think for the LLM-as-Critic stage. We sample 2000 Python seeds in total. One thousand are used to generate Python data, and the remaining one thousand are translated into Swift. This setup fully replicates the AutoCodeGen pipeline.
> > >
> > > **Results**. Following the format of Table 10 in our paper, we list the performance changes across different stages, as shown in below. The results show that after applying Qwen2.5-14B-Instruct for simple problem filtering, models of similar scale, DeepSeek-Coder-V2-Lite, show the largest performance drop. In addition, models from the same family, Qwen-Plus-250718, also show substantial decreases. After applying Qwen-Plus-250718-think as the Critic model, this model achieves the largest performance gain, and another reasoning model, DeepSeek-V3.2-Exp-think, also benefits significantly. These observations are consistent with our hypothesis in Appendix H.
> > >
> > >
> > > |                           | Initial Stage (Rank) | After Simple Problem Filtering (Rank) | After Critic Filtering (Rank) |
> > > | ------------------------- | -------------------- | ------------------------------------- | ----------------------------- |
> > > | DeepSeek-V3.2-Exp think   | 52.1 (2)             | 33.6 (2) -18.5                        | 51.2 (2) +17.6                |
> > > | Qwen-Plus think           | 51.5 (4)             | 33.6 (2)  -17.9                       | 53.1 (1) +19.5                |
> > > | Claude Sonnet 4.5 nothink | 52.4 (1)             | 33.7 (1)  -18.7                       | 51.2 (2) +17.5                |
> > > | Qwen-Plus nothink         | 51.9 (3)             | 33.0 (4)  -18.9                       | 50.0 (4) +17.0                |
> > > | DeepSeek-Coder-V2-Lite    | 44.7 (5)             | 24.5 (5)  -20.2                       | 35.7 (5) +11.2                |
> > >
> > >
> > >
> > > **Case Study**. We observe that the programming problems generated by the Qwen-Plus-250718 exhibit a noticeably different style compared to those generated by the DeepSeek-V3-0324. Problems generated by DeepSeek tend to be more detailed. For example, DeepSeek outputs are usually organized into separate sections, often with subtitles such as “##Problem Description”, “##Input/Output Format”, “##Example Usage”, and “##Notes”. In contrast, problems produced by Qwen typically do not include such subtitles and are usually more concise, often beginning with “Write a ... function named …”. Two illustrative examples are provided below.

---

> > > > ### Author Response · Authors · 2025-11-25
> > > > **Response for Q1 [2/2]**
> > > >
> > > > Two illustrative examples are provided below.
> > > >
> > > >
> > > > A question generated by DeepSeek-V3-0324 (some content is omitted due to length)
> > > >
> > > > ````
> > > > # 3D Shape Analyzer Problem
> > > >
> > > > ## Problem Description
> > > > You are tasked with implementing a `ShapeAnalyzer` class that can analyze 3D shapes represented as 2D grids of cube heights. Each cell in the grid represents a stack of cubes, where the integer value indicates how many cubes are stacked at that position. The class should be able to calculate various properties of the shape including surface area, volume, contact area between adjacent cubes, and bounding box dimensions.
> > > >
> > > > ## Class Requirements
> > > > Implement a class called `ShapeAnalyzer` with the following public member functions:
> > > >
> > > > 1. `int calculateSurfaceArea(const vector<vector<int>>& grid)`
> > > >    - Calculates the total visible surface area of all cubes in the grid
> > > >    - Counts all faces that are not touching another cube or the base
> > > >
> > > > 2. ...
> > > >
> > > > ## Input Format
> > > > All functions take a 2D vector of integers where:
> > > > - Each inner vector represents a row in the grid ...
> > > >
> > > > ## Output Format
> > > > - `calculateSurfaceArea`, `calculateVolume`, and `calculateContactArea` return integers ...
> > > >
> > > > ## Constraints
> > > > - The grid may be empty ...
> > > >
> > > > ## Example Usage
> > > > ```cpp
> > > > ShapeAnalyzer analyzer;
> > > > vector<vector<int>> grid = {{1, 2}, {3, 4}};
> > > >
> > > > int surface = ...
> > > > ```
> > > >
> > > > ## Notes
> > > > - The surface area calculation should account for both external faces and any internal cavities ...
> > > >
> > > > ## Evaluation Criteria
> > > > - Correctness of all calculations ...
> > > > ````
> > > >
> > > > A question generated by Qwen-Plus-250718
> > > >
> > > > ````
> > > > Write a Python function named `maximize_expression` that takes two arguments: a list of integers and a list of arithmetic operators (as strings). The function should return an integer representing the result of applying the operators between the numbers in the order they are given.
> > > >
> > > > The list of numbers will contain one or more integers. The list of operators will contain one fewer element than the list of numbers and may include the following operators: '+', '-', '*'. Apply each operator sequentially from left to right, using the corresponding operands from the number list.
> > > >
> > > > Your task is to compute the final value after evaluating the expression formed by interleaving the numbers and operators in order.
> > > >
> > > > Input:
> > > > - A list of integers `nums`, where 1 <= len(nums) <= 100.
> > > > - A list of strings `ops`, where len(ops) == len(nums) - 1, and each string is one of: '+', '-', '*'.
> > > >
> > > > Output:
> > > > - An integer, the result of evaluating the expression.
> > > >
> > > > Example Usage:
> > > > ```python
> > > > assert maximize_expression([1, 5], ['+']) == 6
> > > > assert maximize_expression([2, 3, 6], ['+', '-']) == -1
> > > > ```
> > > > ````

---

> > > > > ### Author Response · Authors · 2025-11-25
> > > > > **Response for Q2**
> > > > >
> > > > > ## Q2: Multi-Logic task
> > > > >
> > > > >
> > > > > First, we provide a clear definition of multi-logic programming tasks:
> > > > >
> > > > > **Multi-Logic Programming Problem (MLPP)**: MLPP is a programming task whose correct solution requires implementing and coordinating multiple core logical units—such as functions, classes, or modules. Each logical unit corresponds to an independent semantic responsibility or algorithmic objective.
> > > > >
> > > > > We further divide Multi-Logic Problems into two categories:
> > > > >
> > > > > - **Intra-Logic Problems**: multiple logical units collaborate to achieve a single overarching functional goal.
> > > > > - **Inter-Logic Problems**: multiple logical units address distinct functional goals, each independently contributing to the full solution.
> > > > >
> > > > > Second, We found that the main challenges stem from the multiple discrete instructions you mentioned. Numerous and dispersed logical requirements can degrade model performance. On multi-logic tasks, Opus 4.1 shows a 4.0-point drop in performance compared to AutoCodeBench-Full, and on Inter-Logic tasks, its performance decreases by an additional 0.8 points compared with overall multi-logic tasks. In **Appendix L**, we provide a detailed analysis of multi-logic tasks, including their precise definitions and illustrative cases.

---

> ### Author Response · Authors · 2025-11-28
> **A Friendly Reminder for Reviewer 5r4r**
>
> Dear Reviewer 5r4r,
>
> Thank you very much for your careful reading and insightful comments. We have now provided detailed replies to your points on **novelty**, **approximate language translation**, **the clarity of multi-logic task descriptions**, and your suggestion of **re-running AutoCodeGen with a different model family**. We have also added the related clarifications and supplementary analyses to the latest paper.
>
> **If there is anything else we can clarify or further improve, please feel free to reach out anytime. We truly appreciate your time and are more than happy to continue the discussion.**

---

> > ### Comment · Reviewer_5r4r · 2025-11-28
> >
> > Thank you for your detailed rebuttal. After carefully reading your responses to my comments and those of the other reviewers, all my concerns have been addressed.
> >
> > I'm excited to see this contribution to the field of synthetic data generation.
> >
> > As a final suggestion, I found your discussion with Reviewer LmDg regarding the definition order of the I/O format to be very insightful. You might consider incorporating this point into the paper, as I believe other readers would benefit from it as well.

---

> > > ### Author Response · Authors · 2025-11-28
> > >
> > > Dear Reviewer 5r4r,
> > >
> > > Thank you very much for your encouraging follow-up. We truly appreciate that all your earlier concerns have been fully addressed. As suggested, we have incorporated the discussion on the ordering of I/O format definition into the Appendix N, and we agree that this clarification meaningfully improves the exposition of our pipeline.
> > >
> > > Your feedback has significantly strengthened the paper. In light of these revisions and the resolution of all previous concerns, we would kindly ask you to reconsider your score if you feel the updated version better reflects the contribution of the work. We would, of course, be happy to further revise the paper should you see additional opportunities for improvement.
> > >
> > > **Thank you again for your thoughtful evaluation and support throughout the review process**.

---

> > > > ### Comment · Reviewer_5r4r · 2025-11-28
> > > >
> > > > I agree with the authors' explanation and will raise my rating. As a minor suggestion to improve readability, given the extensive length of the appendix, you might consider adding a dedicated Table of Contents for it.

---

### Official Review · Reviewer_LmDg · 2025-10-27

**Soundness:** 2
**Presentation:** 2
**Contribution:** 2
**Rating:** 2
**Confidence:** 4

**Summary:**

This paper proposes AutoCodeGen, a automated pipeline for synthesizing multilingual code generation benchmarks. The key idea is to use LLMs to generate test inputs, execute them in a secure sandbox to obtain ground-truth outputs, and then combine these elements to create verifiable test cases. Based on this pipeline, the authors introduce AutoCodeBench, a benchmark of 3,920 problems spanning 20 programming languages, which they claim is more difficult, diverse, and realistic than existing benchmarks. The paper also releases AutoCodeInstruct, a training dataset. The authors' experiments suggest that current LLMs still struggle significantly on this new benchmark.

**Strengths:**

1. The authors have conducted a comprehensive evaluation across a very large number of models. This provides a broad and valuable snapshot of the current landscape of code generation capabilities as measured by their proposed benchmark.
2. The strategy of generating test inputs first and then using a sandbox to obtain ground-truth outputs is a clever and effective method for ensuring the correctness of the generated test cases.

I am looking forward to the authors' response and would be happy to reconsider my evaluation upon successful clarification.

**Weaknesses:**

The paper's central claim of high difficulty is not sufficiently deconstructed or justified. The methodology for ensuring difficulty involves a post-hoc filtering step that removes any problem solvable by a "moderately capable model" (Line 195). This approach risks conflating genuine, meaningful difficulty with other confounding factors. Specifically, the source of difficulty remains ambiguous:
- Is a problem difficult because it requires complex algorithmic reasoning or deep domain knowledge?
- Is it difficult due to the need to understand obscure language features?
- Or is it "difficult" simply because the problem description is ambiguous, underspecified, or even flawed?

The authors' own finding of a 12.4% error rate in the benchmark (Appendix A) lends significant weight to the third possibility. If a substantial portion of the benchmark is poorly specified, then it may be evaluating a model's ability to "guess the user's intent" rather than its true code generation capabilities. This ambiguity undermines the benchmark's validity as a measure of progress in the field.

**Questions:**

1. The paper reports pass@1 results. Could the authors also provide pass@k results for k > 1 (e.g., k=5, 10)?
2. The paper uses GRPO with the AutoCodeInstruct. What would be the performance if standard Supervised Fine-Tuning (SFT) were used instead? An ablation study comparing GRPO with SFT would help disentangle the benefits of the dataset itself from the specific training strategy employed.
3. The pipeline relies on LLM to generate test inputs. Have the authors considered integrating or comparing this approach with established automated test case generation tools, such as CYaRon or property-based testing libraries (e.g., Hypothesis for Python)?
4. I am slightly confused about the problem generation workflow detailed in Section 2. Specifically, why are the test functions generated before the input/output format specifications? Could you elaborate on how boundary and stress tests are systematically generated within this workflow? It seems that defining the I/O format first would provide a clearer structure for generating comprehensive test cases.

---

> ### Author Response · Authors · 2025-11-25
> **Response for Weakness**
>
> We fully understand the reviewer’s concerns regarding the distinction between genuine difficulty and dataset noise. Automated synthesis inevitably introduces some noise. As the reviewer noted, our human evaluation (Appendix A) indicates a **12.4% error rate**, and Table 7 further shows that even on fully clean samples, **state-of-the-art models still exhibit substantial headroom**, suggesting that difficulty cannot be explained by noise alone.
>
> To more precisely characterize the *source of difficulty* in AutoCodeBench, we conduct a systematic failure-mode analysis on all 1,745 tasks unsolved by Claude Opus 4.1. Using an 8-category checklist (Ambiguous Problem Statement, Unreasonable Test Cases, Algorithmic Logic Error, Missing Functionality, etc.), we asked DeepSeek-V3.2-Exp to label each task. Full details are provided in **Appendix M**. We highlight the key findings:
>
> - **538 / 3920 instances (13.7%)** are identified as low-quality, consistent with our human study (87.6% accuracy). This shows that noise exists but is **limited and quantifiable**.
> - After removing these 538 instances, Opus 4.1 still achieves only **64.3% Pass@1**, confirming that the benchmark remains challenging even in its fully cleaned form.
> - Among the remaining **1,207 high-quality unsolved tasks**, the dominant failure types were **Algorithmic Logic Errors** and **Missing Functionality**. This indicates that the difficulty mainly arises from genuine logical complexity and complex demand.
>
>
> Besides, we fully agree with the reviewer’s insightful comment that ambiguity in problem descriptions may test a model’s ability to infer user intent rather than its code generation ability. However, intent inference is an essential component of real-world programming assistance, where user queries are often incomplete—e.g., “help me implement a quicksort algorithm” typically omits I/O format or boundary conditions. Despite containing a modest amount of such ambiguity (13.7%), **models that perform strongly in practical coding—such as the Claude and OpenAI series—also perform strongly on AutoCodeBench**, suggesting that this ambiguity does not distort the ranking or validity of the benchmark.
>
> In the future, we plan to incorporate our new error analysis method into AutoCodeGen and use stronger models.

---

> > ### Author Response · Authors · 2025-11-25
> > **Response for Q1, Q2, and Q3**
> >
> > ## Q1: Pass@K
> >
> > We have already included pass@k results (k = 1, 3, 5, 7, 9, 10) in Figure 9 and Appendix F of the paper. Specifically, we report the test-time sampling scaling curves for DeepSeek-V3-0324, Qwen3-8B, and Qwen2.5-Coder-32B-Instruct, which clearly show increasing performance as k grows, with diminishing marginal returns.
> > Besides, in Table 4, we believe that using pass@1 is more appropriate for several reasons:
> >
> > - Figure 9 demonstrates that pass@1 and pass@k follow the same performance trend, indicating that pass@1 is a reliable and representative metric.
> > - Our benchmark contains 3,920 problems and 20 programming languages, making it infeasible to report pass@k for every model across all languages.
> > - Prior code generation benchmarks, such as HumanEval, LiveCodeBench, and SWE-Bench[1,2,3], also report pass@1 as the primary metric, and we follow this established convention.
> >
> >
> >
> > > [1] Evaluating Large Language Models Trained on Code, Arxiv21
> > >
> > > [2] LiveCodeBench: Holistic and Contamination Free Evaluation of Large Language Models for Code, ICLR25
> > >
> > > [3] SWE-bench: Can Language Models Resolve Real-World GitHub Issues?, ICLR24
> >
> >
> > ## Q2: AutoCodeInstruct: SFT v.s. GRPO
> > We add the SFT experiments as requested, and the detailed experimental setup and results have been incorporated into **Section 4** of the paper. A summary is provided below:
> >
> > ### **(1) SFT Experimental Setup**
> >
> > For each problem in *AutoCodeInstruct*, we sampled the DeepSeek-V3-0324 model five times and verified the outputs against the test cases to **select one correct solution per problem**. We then used this verified data to fine-tune Qwen2.5-Coder-7B/32B-Instruct. Importantly, we guarantee that **SFT and the two-stage GRPO use exactly the same data**.
> > ### **(2) Results**
> > Unsurprisingly, SFT achieves better performance than GRPO, as it effectively distill knowledge from the stronger teacher model (V3-0324). Meanwhile, **the results from both SFT and GRPO demonstrate the effectiveness and generalizability of *AutoCodeInstruct*.** Detailed experimental results are provided in Table 6.
> > ### **(3) Discussion**
> > While SFT yields better immediate metrics when a strong teacher is available, we highlight RL because a superior teacher is not always accessible (especially for SOTA models trying to improve themselves). The ability to self-improve using verifiable test cases is critical for the long-term evolution of Code LLMs. *AutoCodeInstruct* supports both paradigms effectively.
> >
> > ## Q3: Integrating automated test case generation tools
> > Thank you for raising this insightful question. We have indeed considered integrating traditional automated test-case generation tools, such as CYaRon and property-based testing frameworks like Hypothesis, into the AutoCodeGen pipeline. In fact, CodeI/O[1] adopts a similar idea. However, we ultimately chose the LLM-based approach for the following reasons:
> >
> > > [1] CodeI/O: Condensing Reasoning Patterns via Code Input-Output Prediction, ICML25
> >
> > ### **(1) Multilingual scalability**
> >
> > AutoCodeBench spans **20 programming languages**, while existing automated testing tools (e.g., Hypothesis, QuickCheck, CYaRon) provide **high-quality support only for a few popular languages**. Adopting such tools would require:
> >
> > - Selecting or building one test case generator for each language
> > - Maintaining 20 separate toolchains
> > - Handling differences in language semantics, typing, or environment setup
> >
> > This would make the pipeline **difficult to scale** and inconsistent across languages. In contrast, modern LLMs naturally generalize across languages, enabling a **scalable workflow** for multilingual dataset.
> >
> > ### **(2) Efficiently of the pipeline**
> >
> > To use a tool like Hypothesis for each synthesized code solution, the pipeline would require:
> >
> > 1. Having an LLM synthesize a property-based testing harness,
> > 2. Running the harness to generate large numbers of random test inputs, and
> > 3. Having the LLM convert these sampled inputs into test input function.
> >
> > This introduces significant engineering and computational overhead. By contrast, LLMs can directly generate test input functions.
> >
> > ### **(3) Edge case Coverage**
> >
> > Many test-generation tools rely on *search* or *random mutation*, which often fail to hit edge cases—especially semantic corner cases. Moreover, selecting representative edge cases from a large pool of randomly generated inputs is a challenging issue. LLMs, however, can analyze the code semantics and directly generate targeted edge cases.
> >
> >
> >
> > In summary, while we agree that integrating classical test case generation tools is a promising direction, the LLM-based approach provides a **more scalable, efficient, and semantically aware** solution.

---

> > > ### Author Response · Authors · 2025-11-25
> > > **Response for Q4**
> > >
> > > ## Q4: I/O format
> > > Thank you for pointing out this confusion. We agree that humans often start by defining the I/O format before writing tests. However, in our automated pipeline, generating test inputs first is also valid and offers practical advantages. Below we present the rationale with concrete examples.
> > >
> > > There are two possible workflows:
> > >
> > > 1. **Generate the test inputs first**, and then generate the programming problem (including the I/O format); or
> > > 2. **Generate the I/O format first**, and then generate the test inputs.
> > >
> > > We believe that both approaches are feasible for LLMs, and the former has the additional advantage that the public test cases can be directly embedded into the final problem description, thereby improving clarity and precision.
> > >
> > > Below we illustrate why the first approach is also reasonable.
> > >
> > > Consider the following code solution:
> > >
> > > ```
> > > def cal_two_sum(a, b):
> > >     return a + b
> > > ```
> > >
> > > The model may generate the following test-input function (*our prompt uses explicit specifications and few-shot examples to instruct the model to label corner cases*):
> > >
> > > ```
> > > def private_test_input():
> > >     test_cases = [
> > >         [1, 2],
> > >         [0, 0],                      # boundary test
> > >         [99999999999999999, 3333333333]  # stress test
> > >     ]
> > > ```
> > >
> > > During the problem generation stage, aided by our carefully designed prompt and the model’s strong ability to summarize code behavior, the model may then produce a problem description such as:
> > >
> > > *Write a function named cal_two_sum that computes the sum of two **non-negative integers**.*
> > >
> > > If instead the model generates:
> > >
> > > ```
> > > def private_test_input():
> > >     test_cases = [
> > >         [1, 2],
> > >         [0, 0],
> > >         [4.243, 4.222],   # stress test
> > >         [-0.45, 888],     # boundary test
> > >     ]
> > >
> > > ```
> > >
> > > The resulting problem description may become:
> > >
> > > *Write a function named cal_two_sum that computes the sum of two **floating-point** numbers.*
> > >
> > > In summary,
> > >
> > > - If test cases are generated first, the model infers the I/O format by analyzing the input domain represented in the test cases.
> > > - If the I/O format is specified first, the model is required to construct matching test cases.
> > >
> > > Regardless of which workflow is used, we apply an **LLM-as-Critic stage** to rigorously validate the alignment between the problem description and the test function. Misaligned or ambiguous cases are filtered out.

---

> > > > ### Comment · Reviewer_LmDg · 2025-11-27
> > > >
> > > > I would like to thank the authors for their detailed response, which has fully addressed all of my concerns and provided very reasonable explanations to the questions I raised. I encourage the authors to incorporate these clarifications into the paper, as doing so will result in an even stronger version of the work.
> > > >
> > > > Moreover, the authors’ response and their discussion with the other reviewers have made me realize that automatic benchmark generation is a crucial research direction for efficiently scaling the capabilities of large models. In this regard, the paper offers highly constructive insights and will be very valuable for the design of future benchmarks and evaluation protocols. I am therefore raising my rating and **strongly recommend acceptance**.

---

> > > > ### Comment · Reviewer_LmDg · 2025-11-27
> > > > **A question for future consideration：Repo-Level AutoBench/Dataset？**
> > > >
> > > > Furthermore, I would like to raise a question for future consideration: Do you believe it is feasible to automatically construct repository-level benchmarks or datasets in the future—analogous to tasks such as those presented in SWE-Bench? This seems to represent a significantly more challenging problem setting, but also one with great promise, particularly for the construction of reinforcement learning datasets that could, in turn, drive a virtuous cycle of performance improvement.

---

> > > > > ### Author Response · Authors · 2025-11-28
> > > > > **Response for question: Repo-Level Benchmark/Dataset**
> > > > >
> > > > > We sincerely appreciate your decision to raise the score and for your strong recommendation. We are delighted that our previous response successfully addressed your concerns. We are equally grateful for your inspiring question regarding the future of automated Repo-level benchmarks/datasets.
> > > > >
> > > > > We share your view that automatically constructing Repo-level benchmarks/datasets is not only **feasible** but an **imperative** direction. While recent efforts have explored SWE-style data synthesis[1,2,3,4], a significant bottleneck persists: **the automated construction and diversity guarantee of stateful sandboxes are still below expectation**. We plan to extend the **AutoCodeGen** methodology to the Repo-level to address these challenges. We expect to focus on the following two directions, leveraging the capabilities of Code Agent tools (e.g., Claude Code, Openhands):
> > > > >
> > > > >
> > > > >
> > > > > ### 1.Automated Synthesis of SWE Data
> > > > >
> > > > > We aim to scale the "Reverse Generation" paradigm to full repositories with enhanced verification standards:
> > > > >
> > > > > - **Repo as Seed:** We will use existing open-source repositories as the starting context rather than simple code snippets.
> > > > >
> > > > > - **Per-Instance Stateful Sandbox:** For repo-level tasks, we will move beyond the unified sandbox used in AutoCodeBench by generating a dedicated **stateful sandbox** for each data instance. **Unlike AutoCodeBench, SWE tasks are inherently \*stateful\*, namely the verification relies on the repository and a designated, specific sandbox. Therefore, generating and verifying this executable environment based on the repository seed must be the initial step.** This allows us to construct a unique, stateful environment capable of handling complex project dependencies.
> > > > >
> > > > > - **Dual-State Verification:** We will prompt LLMs to generate "Feature Additions" or "Bug Injections" accompanied by two critical types of tests: **fail-to-pass tests** (to verify the new logic or bug fix) and **pass-to-pass tests** (to ensure zero regression).
> > > > >
> > > > > - **Reverse Issue Generation:** Once a valid pair of `(Code Change, Tests)` is verified in the specific Docker container, we can reverse-synthesize the corresponding GitHub Issue description.
> > > > >
> > > > > ### 2.Automated Synthesis of Terminal-Bench Data
> > > > >
> > > > > Besides the SWE data you mentioned, we believe AutoCodeGen is also uniquely suited for **Terminal-Bench**-like data:
> > > > >
> > > > > - **Environment Synthesis:** Using code snippets or shell tasks as seeds, we will require the LLM to generate a **Dockerfile** to define the required operating system state and pre-installed tools in addition to the solution.
> > > > > - **Stateful Verification:** We will then build and execute these specific Docker environments to verify that the generated scripts interact correctly with the defined OS/Terminal state. Compared to SWE tasks, Terminal tasks are easier to scale because they do not rely heavily on the Repos.
> > > > >
> > > > >
> > > > >
> > > > > We are extremely thankful to the reviewer for raising this question, which provided us with the opportunity to discuss our forward-looking ideas on the Repo-Level direction with you. Furthermore, we have already incorporated several elements discussed during the review process (including the SFT experiments on AutoCodeInstruct, the LLM-based Error Analysis, and the Multi-Logic Task Analysis) into the revised version of our paper, and we have structured this discussion on Repo-level data synthesis into the new **Future Works** section. We thank the reviewer again for the positive and engaging response.
> > > > >
> > > > >
> > > > >
> > > > >
> > > > >
> > > > > > [1] SWE-rebench: An Automated Pipeline for Task Collection and Decontaminated Evaluation of Software Engineering Agents, NeurlPS25
> > > > > >
> > > > > > [2] SWE-smith: Scaling Data for Software Engineering Agents, NeurlPS25
> > > > > >
> > > > > > [3] Training Software Engineering Agents and Verifiers with SWE-Gym, ICML25
> > > > > >
> > > > > > [4] CWM: An Open-Weights LLM for Research on Code Generation with World Models, Arxiv25

---

### Official Review · Reviewer_To2A · 2025-10-31

**Soundness:** 4
**Presentation:** 4
**Contribution:** 4
**Rating:** 6
**Confidence:** 3

**Summary:**

This paper presents an automated framework called AutoCodeGen for constructing high-difficulty, multilingual code generation datasets without manual annotations, which contains 20 programming languages with balanced coverage to rigorously evaluate LLMs on diverse, challenging, and realistic multilingual programming tasks. Experiments reveal that even state-of-the-art models struggle on these tasks, particularly in low-resource languages. The paper also gives complementary training and evaluation resources, including a large-scale, verifiable multilingual training set.

**Strengths:**

1. The automation of the benchmark is good, which eliminates the need for manual annotations. This is a significant advantage for scaling code generation evaluation across languages and problem complexities.

2. AutoCodeBench contains 20 languages and high-difficulty problems, which makes it a robust benchmark for multilingual code generation tasks.

3. The paper is easy to follow.

**Weaknesses:**

1. The authors should pay attention to other low-resource languages.

2. The system heavily relies on LLMs for generating code solutions and test cases, which can introduces biases or errors inherent to the models being used, especially if those models are trained on flawed data.

3. The paper focuses on evaluating the code generation of models but does not address the broader issue of how benchmarks extends to other domains.

**Questions:**

N/A

---

> ### Author Response · Authors · 2025-11-25
> **Response**
>
> We sincerely thank the reviewer for the constructive feedback and for recognizing the strengths of AutoCodeGen and AutoCodeBench. We address each concern in detail below.
>
> ## 1. Weakness1: Low-resource languages
>
> We fully agree that low-resource languages deserve additional attention. **AutoCodeBench already covers 20 programming languages with intentionally balanced distribution, including several low-resource ones such as Racket, Elixir, TypeScript, and Julia**.
>
> Besides, because AutoCodeGen is a fully automated and scalable pipeline, it can naturally extend to more low-resource languages and domain-specific languages (e.g., JSON, XML, DSLs) without manual annotation. We view this automated extensibility as essential for democratizing multilingual code-generation evaluation as models continue to grow in capability.
>
> In the future, we will continue to pay attention to more low-resource programming languages.
>
> ## 2. Weakness2: Potential bias from LLMs
>
> We appreciate the reviewer’s careful point. As acknowledged in the appendix A and H, relying on LLMs for dataset construction may introduce model-specific bias. AutoCodeGen is explicitly designed to mitigate such bias through multiple safeguards:
>
> - **sandbox-verified I/O execution**,
> - **multi-model “push–pull” pipeline** (DeepSeek-V3 for generation, DeepSeek-R1 for critique, DeepSeek-Coder-V2-Lite for filtering),
> - **multi-stage correctness validation** (public/private tests + LLM-as-critic).
>
> While automated generation is not flawless, it offers **unmatched scalability** in an era where model capabilities evolve at rapid pace. Manual annotation cannot keep up with the demand for fresh, diverse, and high-difficulty problems. AutoCodeGen provides a principled way to maintain *continually updated and verifiable* benchmarks.
>
> ## 3. Weakness3: Applicability beyond code generation
>
> AutoCodeBench can be easily extended to the code debugging domain. By sampling real error messages from multiple models and constructing error-driven debugging prompts, we can create scalable and customizable debugging benchmarks tailored to the specific failure patterns of different models. Moreover, the core principle of AutoCodeGen—**obtaining test cases through a verifiable pipeline and deriving problems from solutions and tests**—is highly generalizable, enabling the creation of related tasks such as test case generation benchmarks, code review benchmarks, and many others.

---

### Official Review · Reviewer_6ERu · 2025-11-04

**Soundness:** 2
**Presentation:** 2
**Contribution:** 2
**Rating:** 4
**Confidence:** 3

**Summary:**

Existing benchmarks for evaluating code generation have key limitations: they rely on manual annotation, which is time-consuming and small in scale; they focus on Python code, while benchmarks for other languages have limited difficulty and low coverage.
To address these issues, the authors propose a fully automated framework called AutoCodeGen for building high-difficulty, multi-language code generation datasets. The framework first generates code solutions and test inputs from code snippets, then executes the code solutions in a sandbox to obtain test outputs. Finally, it uses LLMs to combine these three components into test functions, and reversely generates programming problem descriptions based on the solutions and test functions. This process ensures the quality of the benchmark through a three-stage filtering (difficulty, quality, diversity).
Based on this framework, the authors introduce AutoCodeBench, a large-scale benchmark containing 3920 problems across 20 programming languages. Experiments show that even the most advanced models still face significant challenges in the complex and diverse multi-language tasks defined by this benchmark. In addition, the authors also release a supporting training set, AutoCodeInstruct.

**Strengths:**

1. The greatest contribution of this paper is the AutoCodeGen framework, which automatically generates benchmark data and attempts to overcome the reliance on expensive and time-consuming manual annotation. Its highlight lies in ensuring the correctness of the generated data: first, generating test inputs and then obtaining outputs through sandbox execution to avoid potential errors when LLMs directly generate test cases; second, reversely generating problems based on answers and test cases to ensure the actual solvability of the problems.

2. To obtain a high-difficulty and diverse dataset, the paper implements multi-stage filtering. In the problem generation stage, the authors use LLMs as evaluation models to ensure alignment between problem descriptions and test functions (quality control). After generating problem descriptions, they use a "moderately capable" model (DeepSeek-Coder-V2-Lite) to filter out overly simple problems (difficulty control); they also ensure broad data coverage through problem classification and cyclic sampling (diversity control).

3. AutoCodeBench, derived from the AutoCodeGen framework, features high difficulty, multi-language support (20 languages), and balance. Its versions of different sizes also meet the diverse needs of different researchers. Meanwhile, the paper shows that models trained with AutoCodeInstruct (Section 4) have comprehensively improved code generation capabilities, indicating that the data generated by the AutoCodeGen framework is of high quality and generalizability.

**Weaknesses:**

The paper states that AutoCodeBench aims to conduct strict evaluations of large language models on diverse, high-difficulty, and realistic multi-language programming tasks. However, to achieve this goal, there are areas where the paper can be improved:
1. Insufficient Difficulty Assessment in the AutoCodeGen Framework
In Section 2.1.4, the authors mention the method for difficulty control: "we employ a moderately capable code model, DeepSeek-Coder-V2-Lite, to filter out too easy problems. Specifically, we sample answers for each problem ten times using the model and validate the correctness via sandbox execution. We discard problems that are solved in all attempts." This is a filtering method designed to eliminate simple problems, but its limitation is that it is difficult to directly identify high-difficulty problems. The authors later mention in Table 3 how to further distinguish easy/medium/hard problems in the dataset using pass rates. The problem is that while high difficulty equals a low pass rate, a low pass rate does not necessarily mean the problem has real high difficulty (e.g., multi-logic/multi-task). Difficulty assessment based solely on the pass rate indicator requires support from other indicators. In addition, this difficulty assessment imposes requirements on the evaluation model. The paper reports in Section 3.3 that the Pass@1 scores of advanced models on low-resource language problems are relatively higher than those on mainstream languages, which to some extent indicates that the evaluation model is insufficient in filtering simple problems in low-resource languages, leading to differences in the consistency of benchmarks across different languages.
2.  Vague Description of Multi-Logic Tasks in AutoCodeBench
In Section 3.4, the authors mention the concept of multi-logic tasks: "A key feature that distinguishes AutoCodeBench from prior benchmarks is the inclusion of multilogical problems. These problems require models to implement multiple distinct functions or classes within a single task, challenging their ability to handle multiple core demands simultaneously." In subsequent experiments, the authors show that advanced models perform poorly on such tasks. As a representative of high-difficulty problems, multi-logic tasks can indeed reflect a model's comprehensive ability to handle complex code tasks. However, the description of multi-logic tasks in the paper is somewhat brief, and such problems are not generated through active control; the analysis of such problems is more inclined to data mining.

**Questions:**

1. (Regarding Weakness 1) Given that the relatively simple difficulty filter is insufficient in distinguishing high-difficulty problems and filtering simple problems in low-resource languages, have you considered adopting improved methods? For example, could you use a more capable model as an evaluator to try more reasonable filtering for problems in different languages? Or, similar to quality control, could you incorporate more LLM-based analysis (e.g., using LLMs to analyze unsolved problems and solutions) to assist in filtering truly high-difficulty problems?
2. (Regarding Weakness 2) Could you supplement more detailed analysis of multi-logic tasks? Could you provide more detailed statistics on the complexity of these problems? For instance, how many functions do they require to implement on average? Is there a specific reflection of the low pass rate of advanced models on high-difficulty problems?

---

> ### Author Response · Authors · 2025-11-25
> **Response for W1 and Q1**
>
> We sincerely appreciate the reviewer’s thoughtful observations regarding the limitations of pass-rate–based difficulty estimation and the potential multilingual bias introduced by the DeepSeek-Coder-V2-Lite. We agree that pass rates alone cannot fully capture human-perceived difficulty, nor can a single filtering model perfectly normalize difficulty across languages. However, we believe this does not affect AutoCodeBench’s validity as a benchmark.
>
> First, AutoCodeBench still is challenging and reliable differentiates. As shown in Table 4, the state-of-the-art model Claude Opus 4.1 achieves only a 55.4 average Pass@1. Furthermore, there is a 47.5 gap between the best and worst models. This evidence confirms that AutoCodeBench is a **high-difficulty, valid benchmark** that **powerfully differentiates model capabilities**.
>
> Second, we still believe that the pass rate is a valuable difficulty metric, a method widely adopted by many data synthesis works[1,2,3]. We emphasize that **model-perceived difficulty** often diverges from **human-perceived difficulty**. Models occasionally fail on tasks that humans deem very simple. Incorporating these specific failure cases into the benchmark is highly valuable, and the pass rate effectively helps us achieve this goal. Therefore, we believe that assessing difficulty from a **model-perceived perspective (pass rate)** is reasonable, a practice that is also common in the field of code benchmark construction[4,5].
>
> Moving forward, we intend to follow the reviewer’s excellent suggestion. We will replace DeepSeek-Coder-V2-Lite with a stronger and more balanced filtering model and complement this approach with LLM-based difficulty estimation methods in the AutoCodeGen.
>
>
>
> > [1] DART-Math: Difficulty-Aware Rejection Tuning for Mathematical Problem-Solving, NeurlPS24
> >
> > [2] Unleashing LLM Reasoning Capability via Scalable Question Synthesis from Scratch, ACL25
> >
> > [3] QueST: Incentivizing LLMs to Generate Difficult Problems, Arxiv25
> >
> > [4] FullStack Bench: Evaluating LLMs as Full Stack Coders, Arxiv24
> >
> > [5] [o1 tops aider’s new polyglot leaderboard | aider](https://aider.chat/2024/12/21/polyglot.html)
>
>
>
> ## Difficulty Analysis
>
> During the rebuttal period, we add Appendix L (Multi-Logic Task Analysis) and Appendix M (LLM-based Error Analysis) in the paper to further analyze the quality and difficulty characteristics of AutoCodeBench. Below we summarize the key findings.
>
> First, in Appendix M, we design a checklist covering both quality issues and difficulty-related error types, and apply DeepSeek-V3.2-Exp to analyze the 1,745 tasks that Claude Opus 4.1 failed to solve. After removing noisy samples such as ambiguous problem statements or unreasonable test cases, we find that among the remaining high-quality, high-difficulty tasks, the dominant failure modes are *Algorithmic Logic Errors* and *Missing Functionality*. This indicates that the difficulty mainly arises from genuine logical complexity and complex demand. Moreover, even after excluding all low-quality noisy samples, Opus 4.1 still achieves only **64.3%**, demonstrating that the remaining tasks remain substantially challenging.
>
> Second, in Appendix L, we categorize multi-logic tasks into two types:
>
> 1. **Intra-Logic Problems**, where multiple logical units collaborate to achieve a single overarching functional objective; and
> 2. **Inter-Logic Problems**, where multiple logical units correspond to distinct functional objectives that independently contribute to the full solution.
>
> We then use DeepSeek-V3.2-Exp to compute the number of logical units per task and the distribution between Inter-Logic and Intra-Logic categories. On average, each problem contains **3.37 logical units**. Among them, **389** are Inter-Logic, **1,223** are Intra-Logic, and **10** exhibit characteristics of both types. Claude Opus 4.1 achieves **50.6%** and **51.5%** on the Inter-Logic and Intra-Logic categories, respectively. These results suggest the difficulty increases when the logical units are functionally separate and do not work toward the same objective.. We present two illustrative case studies in Figures 18 and 19 of the paper.

---

> ### Author Response · Authors · 2025-11-25
> **Response for W2 and Q2**
>
> Thank you for pointing out that the description of multi-logic tasks was not sufficiently detailed. We offer the following clarifications.
>
> Before running AutoCodeGen, we found that many code snippets naturally contain **multiple functions or classes**—a common characteristic of real-world repos. To preserve this realism and maintain the fully automated, human-free nature of our pipeline, we **intentionally did not decompose** such snippets into single-function units. As a result, multi-logic tasks emerge naturally in the generated benchmark, reflecting real programming scenarios rather than artifactually simplified ones.
>
> In the revised version of the paper, we have **added a comprehensive analysis in Appendix L**, where we describe in detail the data statistics process for identifying multi-logic tasks and provide concrete case examples. Here are some key points.
>
> We provide a clear definition of multi-logic programming tasks:
>
> - **Multi-Logic Programming Problem (MLPP)**: MLPP is a programming task whose correct solution requires implementing and coordinating multiple core logical units—such as functions, classes, or modules. Each logical unit corresponds to an independent semantic responsibility or algorithmic objective.
>
> As mentioned earlier in the Response for Weak1 and Q1 section, we further divide Multi-Logic Problems into two categories: Intra-Logic Problems and Inter-Logic Problems.
>
> The brief statistics is shown in below:
>
> - **Problem description length**: 576.4 (vs. 498.2 overall)
> - **Solution length**: 610.0 (vs. 487.5 overall)
> - **Average number of logical components** (functions/classes): **3.37**
>
> These statistics demonstrate that multi-logic tasks are substantially more complex than typical single-function problems.

---

> ### Author Response · Authors · 2025-11-28
> **A Friendly Reminder for Reviewer 6ERu**
>
> Dear Reviewer 6ERu,
>
> Thank you again for your thoughtful and constructive comments. We wanted to kindly let you know that we have already provided detailed responses to your questions regarding the **difficulty assessment** and the **clarity of multi-logic task descriptions** over the past two days. We have also incorporated the corresponding clarifications and additional analyses into the updated version of the paper (see Appendix).
>
> **If you have any further questions or would like us to elaborate on any part, please feel free to reach out anytime. We would be very happy to continue the discussion and provide any clarification that may help.**

---

### Author Response · Authors · 2025-11-25
**Response for all Reviewers**

**We would like to thank all reviewers for their valuable suggestions and questions**. We have provided detailed responses to each of the identified weaknesses and questions. In summary, during the rebuttal period, we conducted the following additional analyses and clarifications:

1. Provided a more detailed analysis of the multi-logic tasks unique to our work, in response to Reviewers 6ERu and 5r4r.
2. Conducted a more thorough quality analysis and error-type analysis, in response to Reviewers 6ERu and LmDg.
3. Analyzed the advantages and limitations of using the pass rate for difficulty estimation, in response to Reviewer 6ERu.
4. Clarified the importance AutoCodeBench places on low-resource languages, in response to Reviewer To2A.
5. Explained the model bias issue inherent in automated data synthesis, in response to Reviewer To2A.
6. Clarified the general applicability of the AutoCodeGen method, in response to Reviewer To2A.
7. Compared Pass@1 and Pass@k metrics, in response to Reviewer LmDg.
8. Added new SFT experiments on AutoCodeInstruct, in response to Reviewer LmDg.
9. Compared automated test-case generation tools with LLM-based generation, in response to Reviewer LmDg.
10. Analyzed the sequencing of test-case generation and I/O format generation, again in response to Reviewer LmDg.
11. Highlighted the differences between our work and prior approaches, in response to Reviewer 5r4r.
12. Added a small-scale approximate language translation experiment, in response to Reviewer 5r4r.
13. Added an experiment running the entire AutoCodeGen pipeline with the Qwen model family, in response to Reviewer 5r4r.

Several clarifications and analyses have also been incorporated into the revised manuscript, and **all newly added content is highlighted in red**.

**In addition, we appreciate the reviewers’ recognition of the novelty and practicality of our AutoCodeGen framework, as well as the community value of the AutoCodeBench benchmark. We hope our responses address your concerns, and we look forward to further discussion and feedback.**

---

### Author Response · Authors · 2025-11-27
**General Response**

Dear Reviewers,

We have submitted our rebuttal and provided detailed responses to all weaknesses and questions. If any clarification is still needed, we would be happy to provide additional explanations. We look forward to any further feedback you may have.

---

### Author Response · Authors · 2025-12-03
**Summary of Rebuttal [1/2]**

Dear AC,

Thank you very much for your time and for overseeing the review process of our work. Due to an OpenReview bug that interrupted the rebuttal process, we would like to provide a consolidated summary here to help you clearly understand the full scope of our rebuttal interactions.

Overall, we have had in-depth and productive discussions with **Reviewer LmDg** and **Reviewer 5r4r**. Notably, for **Reviewer LmDg**, although their initial rating was 2, they explicitly stated in the original review:

*“I am looking forward to the authors’ response and would be happy to reconsider my evaluation upon successful clarification.”*

After we provided extensive analytical experiments and addressed all five of their concerns in detail, the reviewer responded positively by **raising their score and recommending acceptance**. As this change directly aligns with their stated condition, we believe the updated score is reliable.

Similarly, we conducted multiple rounds of discussion with **Reviewer 5r4r**, successfully resolving their concerns as well. This reviewer also provided **positive feedback and indicated that they would raise their score**. For the remaining two reviewers, we have provided thorough point-by-point responses addressing all of their comments.

Below is a detailed summary of the rebuttal.

## **Strengths Recognized by the Reviewers**
- **Automated benchmark/dataset construction** (core innovation)
    - Acknowledged by Reviewers 6ERu, To2A, and 5r4r for being human-free, scalable, and efficient.
- **Three-step interactive test generation with LLM-Sandbox** (core innovation)
    - Highlighted by Reviewers 6ERu and LmDg.
- **Robust, multilingual evaluation of a wide range of models** (core contribution)
    - Praised by Reviewers To2A and LmDg for demonstrating strong generality.
- **Full open-sourcing of AutoCodeBench, AutoCodeGen, AutoCodeInstruct, and Sandbox** (core contribution)
     - Appreciated by Reviewers 6ERu and 5r4r.
- **Multi-stage filtering (difficulty, quality, diversity)**
   - Identified as a methodological strength by Reviewer 6ERu.
- **High quality and strong generalization of LLM-generated data**
    - Noted by Reviewer 6ERu.
- **Paper clarity and readability**
    - Highlighted by Reviewer To2A.

## **Concerns Raised by Reviewers & Our Responses**

**1. Difficulty and quality evaluation (6ERu / LmDg)**

- Reviewer 6ERu argued that a low pass@k does not necessarily indicate true difficulty. We emphasize that **model-perceived difficulty** often diverges from **human-perceived difficulty**. pass@k effectively identifies problems that are challenging *from the model’s perspective*, which is precisely what we need when constructing a benchmark designed to evaluate LLMs.
- Regarding the cross-language difficulty imbalance raised by Reviewer 6ERu, we acknowledge that this is partly due to the limited capability of the model used for filtering and will use stronger models in future updates. We also point out that AutoCodeBench evaluates *the overall capability of different models*, rather than to assess one model’s performance *across different programming languages*. Therefore, some degree of difficulty imbalance across languages is acceptable.
- Reviewer LmDg suspected that the difficulty may come from vague descriptions. Our additional analyses and experiments in Appendix M confirm that the difficulty is genuine rather than caused by ambiguity.

**2. Insufficient description of multi-logic tasks (6ERu / 5r4r)**

- We provided additional statistics and analyses, now included in Appendix L.

**3. Extending to more languages (To2A)**

- We clarified the extensibility of AutoCodeGen and expressed plans to include more low-resource languages.

**4. LLM bias concerns (To2A)**

- We reiterated our discussions in Appendices A and H and explained how bias is mitigated.

**5. Extension to other domains (To2A)**

- We emphasized that AutoCodeGen can generalize easily and provided code debug task as a concrete example.

**6. Pass@k results (LmDg)**

- We pointed to Appendix F where pass@k is already provided and explained why the main table focuses on pass@1.

**7. SFT experiments (LmDg)**

- We added SFT results for Qwen2.5-Coder-7B/32B-Instruct, demonstrating strong effectiveness and generalizability of AutoCodeInstruct (now in Section 4 and Appendix I).

**8. Automated test-generation tools vs LLM-based (LmDg)**

- We clarified key differences and justified our choice based on multilingual scalability, pipeline efficiency, and edge-case coverage.

**9. Ordering of I/O format vs test function generation (LmDg)**

- We analyzed both orderings and explained that LLMs handle either direction, and clarified why our chosen order is preferable (now in Appendix N).

**10. Distinction from prior data synthesis methods (5r4r)**

- We clarified that AutoCodeGen focuses on multilingual, verifiable data with a unique three-step test-case generation pipeline, unlike Evol-Instruct/OSS-Instruct.

---

> ### Author Response · Authors · 2025-12-03
> **Summary of Rebuttal [2/2]**
>
> **11. "Naturalness" of translated data (5r4r)**
>
> - We conducted a new experiment on Swift showing comparable levels of “Swifty-ness” between translated and native data.
>
> **12. Running AutoCodeGen with different model families (5r4r)**
>
> - We added experiments using Qwen models and included case studies, showing stylistic differences with DeepSeek (Qwen tends to produce more concise problem statements).
>
>
>
> ## Newly Added Content in the Revised Paper
>
> All newly added content in the rebuttal has been highlighted in **red** within the revised paper. These additions include:
>
> - **Section 4 and Appendix I**: SFT experiments for AutoCodeInstruct;
>
> - **Section 6**: Expanded discussion of future work, including the automated construction of **repo-level benchmarks/datasets**;
>
> - **Appendix L**: Formal definitions, data analysis, and case studies for **Multi-Logic Tasks**;
>
> - **Appendix M**: **LLM-based error analysis**, including detailed difficulty analysis;
>
> - **Appendix N**: Clarification regarding the ordering of **I/O Format** generation.

---

### Meta-Review · Area_Chair_7uxK · 2025-12-06

**Summary:**

1. Difficulty and quality evaluation (6ERu / LmDg)
2. Insufficient description of multi-logic tasks (6ERu / 5r4r)
3. Extending to more languages (To2A)
4. LLM bias concerns (To2A)
5. Extension to other domains (To2A)
6. Pass@k results (LmDg)
7. SFT experiments (LmDg)
8. Automated test-generation tools vs LLM-based (LmDg)
9. Ordering of I/O format vs test function generation (LmDg)
10. Distinction from prior data synthesis methods (5r4r)

**Reviewer Concerns:**

All the reviewer concerns listed above were addressed by the rebuttal.

**Reviewer Scores:**

As the concerns are well addressed by the rebuttal, I guess:
- Reviewer 6ERu will raise the score from 4 to 6.
- Reviewer To2A will keep the positive score of 6.
- Reviewer LmDg will raise the score from 2 to 6.
- Reviewer 5r4r will raise the score from 4 to 6.

---

### Decision · Program_Chairs · 2026-01-26

Accept (Poster)